# Multimode ultrastrong coupling in three-dimensional photonic-crystal cavities

Fuyang Tay [1,2], Ali Mojibpour[1], Stephen Sanders [1], Shuang Liang [3,4], Hongjing Xu [5], Geoff C. Gardner [6], Andrey Baydin [1,7,8], Michael J. Manfra [3,4,6,9], Alessandro Alabastri[1,7,8], David Hagenmüller [10] & Junichiro Kono [1,5,7,8,11] ✉

Recent theoretical studies have highlighted how spatially varying cavity electromagnetic fields enable novel cavity quantum electrodynamics phenomena, such as the Dicke superradiant phase transition. Three-dimensional photonic-crystal cavities, which exhibit discrete in-plane translational symmetry, overcome this limitation, but fabrication challenges have hindered the achievement of strong coupling. Here, we demonstrate multimode ultrastrong coupling between cavity modes of a three-dimensional photonic-crystal cavity at terahertz frequencies and the cyclotron resonance of a Landau-quantized two-dimensional electron gas in gallium arsenide. The multimode coupling depends on the spatial profiles of the cavity modes, resulting in distinct coupling scenarios based on probe polarization. Our results align with an extended multimode Hopfield model that accounts for spatial field variations. Guided by the model, we discuss possible strong ground-state correlations between cavity modes and introduce relevant figures of merit for multimode ultrastrong coupling. Our findings highlight the crucial role of spatial inhomogeneity in multimode ultrastrong coupling.

The interaction between a two-level system and a spatially uniform cavity electric field represents the most fundamental model in the field of cavity quantum electrodynamics (cQED). This model has been extensively studied over the past century and has provided profound insights into the nature of light–matter interactions in confined geometries. Recently, significant phenomena have been predicted to emerge when the cavity electromagnetic field is spatially nonuniform[1–3]. For example, the Dicke superradiant phase transition (SRPT)[4,5], which is expected to occur when the light–matter coupling strength is comparable to the photon frequency but prohibited by a no-go theorem in the case of a uniform field[6,7], may occur when a spatially varying electromagnetic field is involved. The Dicke SRPT

becomes possible when the cavity field strongly couples with a paramagnetic instability at finite wave vectors, which is only possible with a spatially varying electric field[1–3,8].

Fabry–Pérot cavities with Bragg mirrors, or one-dimensional photonic-crystal cavities (1D-PCCs), are commonly employed to investigate solid-state cQED with quantum wells[9–12]. These purely dielectric cavities offer a significant advantage in terms of high-quality factors[10–12]. However, the cavity mode profile in 1D-PCCs is spatially uniform in the plane perpendicular to the stacking direction. Three-dimensional (3D) PCCs overcome this limitation by exhibiting discrete in-plane translational invariance[13–15]. This allows cavity photons to couple with matter excitations with a set of in-plane reciprocal lattice

[1]Department of Electrical and Computer Engineering, Rice University, Houston, TX, USA. [2]Applied Physics Graduate Program, Smalley-Curl Institute, Rice University, Houston, TX, USA. [3]Department of Physics and Astronomy, Purdue University, West Lafayette, IN, USA. [4]Birck Nanotechnology Center, Purdue University, West Lafayette, IN, USA. [5]Department of Physics and Astronomy, Rice University, Houston, TX, USA. [6]School of Electrical and Computer Engineering, Purdue University, West Lafayette, IN, USA. [7]Smalley-Curl Institute, Rice University, Houston, TX, USA. [8]Rice Advanced Materials Institute, Rice University, Houston, TX, USA. [9]School of Materials Engineering, Purdue University, West Lafayette, IN, USA. [10]CESQ-ISIS (UMR 7006), Université de Strasbourg and CNRS, Strasbourg, France. [11]Department of Materials Science and NanoEngineering, Rice University, Houston, TX, USA. ✉e-mail: kono@rice.edu

vectors. Coherent radiative hopping and significant changes in exciton dispersion have been predicted to occur in 3D-PCCs operating in the strong coupling regime[16]. Nevertheless, fabrication challenges in 3D-PCCs have prevented the realization of extreme regimes of light–matter interactions, which necessitate high-quality factors and strong electric field confinement[17]. This includes the strong and the ultrastrong coupling[18,19] (USC) regimes, the latter being achieved when the light–matter coupling strength is a non-negligible fraction of the bare resonance frequencies.

In the present work, we demonstrate multimode USC in a terahertz (THz) 3D-PCC. The cavity modes of the 3D-PCC are simultaneously coupled to the cyclotron resonance (CR) of an ultrahigh-mobility two-dimensional electron gas (2DEG) in GaAs. Our 3D-PCC, which is nearly inversion-symmetric in the stacking direction, exhibits two degenerate cavity modes with different spatial profiles in the 2DEG plane depending on their polarization. Although the dipole approximation remains valid (the spatial profiles of the cavity modes are nearly uniform on the scale of the electron cyclotron orbits), the spatial variation of the cavity field within a unit cell of the 3D-PCC significantly affects the hybridization between the cavity modes through the CR, leading to the emergence of novel physics. Two distinct multimode scenarios in which different cavity modes are either coupled or decoupled via the CR are observed in a single device, by

rotating the polarization of the probe. Our experimental results show excellent agreement with numerical simulations and calculations based on an extended Hopfield model, which takes into account the spatial variation of the cavity field. We discuss possible correlations between the cavity modes in the ground state of the system as indicated by the model and present relevant figures of merit for multimode systems with USC. This work provides an approach to explore light–matter interactions beyond the weak coupling regime in 3D-PCCs and highlights the importance of spatial variation of cavity mode profiles in cQED.

## Results

### Polarization-dependent multimode coupling in a 3D-PCC

3D photonic crystals are dielectric structures that exhibit periodic refractive index modulations in all three dimensions. Here, we consider the woodpile structure where the rod arrays are stacked in an alternating orthogonal pattern, which is known to exhibit a complete photonic band gap[20,21]. We used photolithography and deep reactive ion etching to fabricate a rod array in silicon wafers (Fig. 1a, Supplementary Fig. 9, see Methods for more details). A woodpile lattice[15,22] is formed by stacking patterned silicon wafers sequentially (Fig. 1b). To induce photonic modes in the photonic band gap, a planar defect (a 60-μm-thick GaAs layer) was inserted at the center of the woodpile

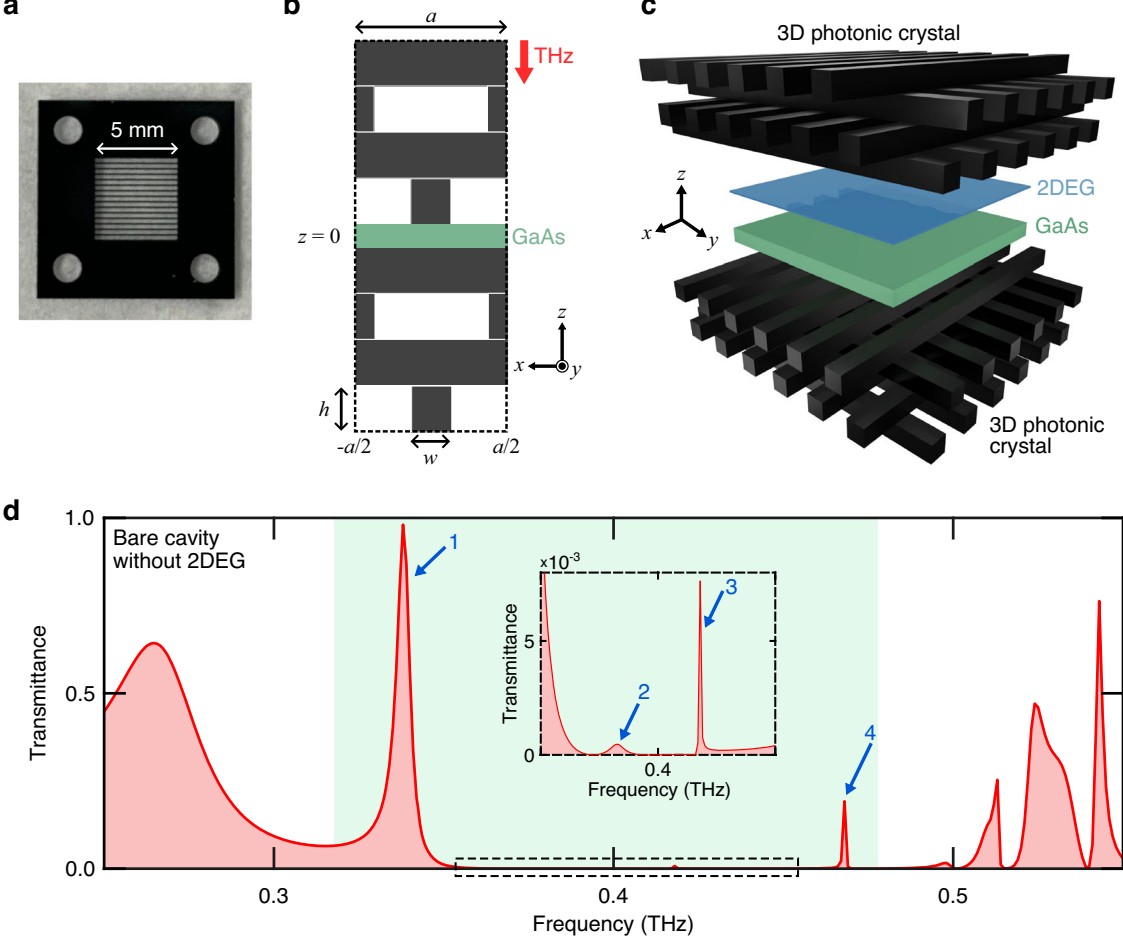

**Fig. 1 | The 3D-PCC. a** A photo of a fabricated architecture containing the designed woodpile structure. **b** Side view of the unit cell of the 3D-PCC. A GaAs wafer is sandwiched by a pair of 3D photonic crystals, which consists of four layers with a rod array in the $z$-direction. Each silicon rod had a width of $w = 0.26a$ and a height of $h = 0.3a$ where $a = 333$ μm is the lattice constant. The rods in each successive layer of the 3D photonic crystal are rotated by 90°. The rods in the third and fourth layers are displaced by $0.5a$ relative to the rods in the first and second layers, respectively. The THz radiation propagates along the $-z$-direction. **c** Schematic of a 2DEG embedded in a 3D woodpile cavity. **d** Transmittance spectrum of the bare 3D-PCC from numerical simulations. The shaded region corresponds to the photonic band gap computed from the band structure of an infinite woodpile lattice (in all directions) without defect. The blue arrows mark the doubly degenerate cavity modes (with respect to the two polarizations) that are labeled by $p = 1, 2, 3, 4$ throughout the paper. The inset is a magnified view of the spectrum.

lattice, which breaks discrete translational invariance in the stacking direction $z$ (Fig. 1b). We performed numerical simulations with COMSOL Multiphysics to calculate the transmittance spectrum of the woodpile structure under normal incidence, as shown in Fig. 1d. To maintain high peak amplitude, only four silicon layers were placed on each side of the GaAs defect layer. Four cavity modes can be observed within the photonic band gap. Those cavity modes exhibit finite quality factors that depend on their position in the band gap: The closer they are to the center of the band gap, the larger their quality factor is. Specifically, the quality factors of the modes with frequencies $\omega_{p=1}/2\pi = 338$ GHz, $\omega_{p=2}/2\pi = 382$ GHz, $\omega_{p=3}/2\pi = 417$ GHz, and $\omega_{p=4}/2\pi = 468$ GHz, as computed by finite difference time-domain (FDTD) simulations, are 72, 70, 6200, and 1540, respectively.

The woodpile 3D-PCC exhibits four mirror symmetry planes per unit cell in the $xy$ plane. In the central unit cell, those mirror symmetry planes are $x = 0$; $\pm a/2$ and $y = 0$; $\pm a/2$ (see Fig. 1b). Depending on the incident polarization, $\sigma$, two families of cavity modes can be distinguished due to these symmetries. The modes excited by THz light polarized along $x$ are even (odd) with respect to $y = 0$; $\pm a/2$ ($x = 0$; $\pm a/2$). These modes are mostly linearly polarized in the $xz$-plane and are referred to as $\sigma = x$. On the other hand, the modes excited by THz light polarized along $y$ are odd (even) with respect to $y = 0$; $\pm a/2$ ($x = 0$; $\pm a/2$). These modes are mostly linearly polarized in the $yz$-plane and are referred to as $\sigma = y$. Furthermore, the 3D-PCC features another symmetry: a combination of a 4-fold rotational symmetry with respect to the $z$-axis and an inversion symmetry with respect to the plane $z = 0$. This symmetry results in the degenerate frequency for the $\sigma = x$ and $\sigma = y$ modes in the spectrum[23] (see Supplementary Figs. 10–12 and Supplementary Section 3). However, the lack of inversion symmetry leads to asymmetric cavity profiles along the $z$-direction. For instance, Fig. 2a, b display the expectation value of the electric energy density in the vacuum state $|0\rangle$, $\langle 0|I_{p,\sigma}(z)|0\rangle$, for the cavity modes $p = 1$ with $\sigma = x$ and $\sigma = y$. Therefore, despite having the same resonance frequency, the cavity modes excited by orthogonal polarizations exhibit different spatial profiles. Importantly, this structure is crucial in investigating the role of cavity mode profiles in multimode coupling because the cavity modes excited by orthogonal polarizations exhibit the same resonance frequency and comparable light–matter coupling strengths, with their primary distinction being the spatial profiles. It should be noted that all cavity modes are localized in the vicinity of the defect GaAs layer with a standard deviation below one Si rod thick for $p = 3, 4$ and about two logs thick for $p = 1, 2$ (see Supplementary Section 1).

To study the coupling between the cavity modes of the 3D-PCC and the CR of a 2DEG in GaAs, with dipole moment in the $xy$-plane, the GaAs defect layer is replaced by a GaAs substrate containing a multiple quantum well heterostructure near the top surface. The total thickness of the sample remains at around 60 μm. The standard deviation of the in-plane electric field in the electromagnetic vacuum state, $\sqrt{\langle 0|E_{p,\sigma}^2(\boldsymbol{\rho})|0\rangle}$ (see Supplementary Section 1 for definition), for the first two bare cavity modes $p = 1, 2$ and $\sigma = x, y$ is displayed in Fig. 2c–f at the vertical location of the 2DEG, $z_{2\text{DEG}}$. For $\sigma = x$, the cavity electric field is tightly confined at the edges of the rods (Fig. 2c, d). Conversely, the mode profiles for $\sigma = y$ distribute more evenly within the unit cell (Fig. 2e, f). It should be noted that while the two families of modes $\sigma = x, y$ provide a good way to classify the cavity modes without 2DEG, those two polarizations are coupled by the off-diagonal elements of the dielectric tensor describing the CR (see Supplementary Section 2). However, it turns out that such a coupling is small and the polariton eigenmodes can thus still be well identified by their polarization index $\sigma$, as shown below.

We conducted numerical simulations with COMSOL Multiphysics for the 3D-PCC with a GaAs 2DEG under an external magnetic field, $B$, along the $z$-direction (Supplementary Section 2). Figure 3a, b shows transmittance spectra for the $x$ and $y$ polarizations, respectively. For

simplicity, we restrict ourselves to the first two modes $p = 1, 2$ in the following discussion, since the modes $p = 3, 4$ with extremely narrow line widths were not observed in the experimental data due to the limited resolution of THz spectroscopy. The cyclotron frequency, $\omega_c = eB/m_{\text{eff}}$, is directly proportional to $B$ (Supplementary Fig. 8). Four polariton branches, two upper polaritons (UPs) and two lower polaritons (LPs), are observed for the $x$ polarization (Fig. 3a), accompanied by a splitting spanning the space around the line of the bare CR frequency (white dashed), $\omega_c$, that separates the UPs and LPs. By contrast, this splitting is not visible for the $y$ polarization (Fig. 3b). Instead, we observed an S-shaped polariton branch that crosses the $\omega_c$ line and behaves as an LP for cavity mode $p = 2$ at high $B$. The broad transmittance background at low frequencies is the photonic band edge, as shown in Fig. 1d.

We performed THz time-domain spectroscopy measurements to obtain the transmittance spectra under the same conditions. All measurements were taken at 4 K in a cryostat with a superconducting magnet. As the measurements directly collected the transmitted THz waveform in the time domain, the frequency resolution of the data is limited by the length of the time-domain traces that were used in the Fourier transformation. Note that the optical components in the system, for example, cryostat windows and crystals for THz generation and detection, induced echoes of the THz pulses at longer time delays. Including the echoes in the Fourier transformation causes artificial dips in the spectrum (the Fabry–Pérot effect)[12]. Therefore, the time-domain traces used to plot the color plots in Fig. 3c, d were truncated before the arrival of THz echoes (33 ps) and zero padded to avoid the artificial dips. The frequency resolution of the color plot was limited. Furthermore, the white dots denote the $B$-dependent polariton frequencies that were extracted from longer time-domain traces through fittings. More details about the differences between short and long time-domain traces (Supplementary Figs. 5 and 13) and the extraction procedures (Supplementary Fig. 15) are discussed in Supplementary Section 3. The experimental results show good agreement with the simulations, namely, we observed a polariton splitting that separates the UPs and LPs for the $x$ polarization and no splitting for the $y$ polarization. The UPs at high frequencies were not observed in experiments likely because the transmission is low for resonances near the center of the photonic band gap and also because there exist additional losses induced by imperfections in the fabricated woodpile lattice.

## Theoretical analysis

To gain a better understanding of the observed polarization-dependent transmittance spectra, we developed a microscopic quantum model described by the Hamiltonian $\hat{H} = \hat{H}_{\text{cav}} + \hat{H}_{\text{CR}} + \hat{H}_{\text{int}} + \hat{H}_{A^2}$ (Supplementary Section 1). The free-photon Hamiltonian is $\hat{H}_{\text{cav}} = \sum_{p,\sigma} \hbar\omega_p \hat{a}_{p,\sigma}^\dagger \hat{a}_{p,\sigma}$, with $\hat{a}_{p,\sigma}$ the annihilation operator of a photon in mode $p$ with polarization $\sigma$ and frequency $\omega_p$. The effective CR Hamiltonian $\hat{H}_{\text{CR}} = \hbar\omega_c \int \frac{d\boldsymbol{\rho}}{a} \hat{b}^\dagger(\boldsymbol{\rho})\hat{b}(\boldsymbol{\rho})$ is written in terms of the collective CR operators at the in-plane position $\boldsymbol{\rho}$ in the woodpile unit cell. The associated creation operator,

$$\hat{b}^\dagger(\boldsymbol{\rho}) = \frac{1}{a\sqrt{\mathcal{N}}} \sum_{k,\mathbf{G}} \hat{c}_{\nu,k-G_y}^\dagger \hat{c}_{\nu-1,k} e^{iG_x k l_c^2} e^{-i\mathbf{G}\cdot\boldsymbol{\rho}}, \quad (1)$$

promotes an electron from the highest-occupied Landau level (LL) $\nu-1$ with momentum $k$ to the lowest-unoccupied LL $\nu$ and momentum $k-G_y$, with $\hat{c}$ and $\hat{c}^\dagger$ the electron annihilation and creation operators, $G_j = m_j \times 2\pi/a$ ($m_j = 0, 1, 2, \ldots$) the $j = x, y$ component of the in-plane reciprocal lattice vector of the 3D-PCC, $l_c = \sqrt{\hbar/eB}$ the magnetic length, $e$ the electron charge, $\nu$ the LL filling factor, and $\mathcal{N}$ the LL degeneracy (see Supplementary Section 1). The collective CR operators approximately satisfy bosonic commutation relations $[\hat{b}(\boldsymbol{\rho}), \hat{b}^\dagger(\boldsymbol{\rho}')] = \delta(\boldsymbol{\rho} - \boldsymbol{\rho}')$ in the dilute regime. We emphasize that those degenerate CR modes

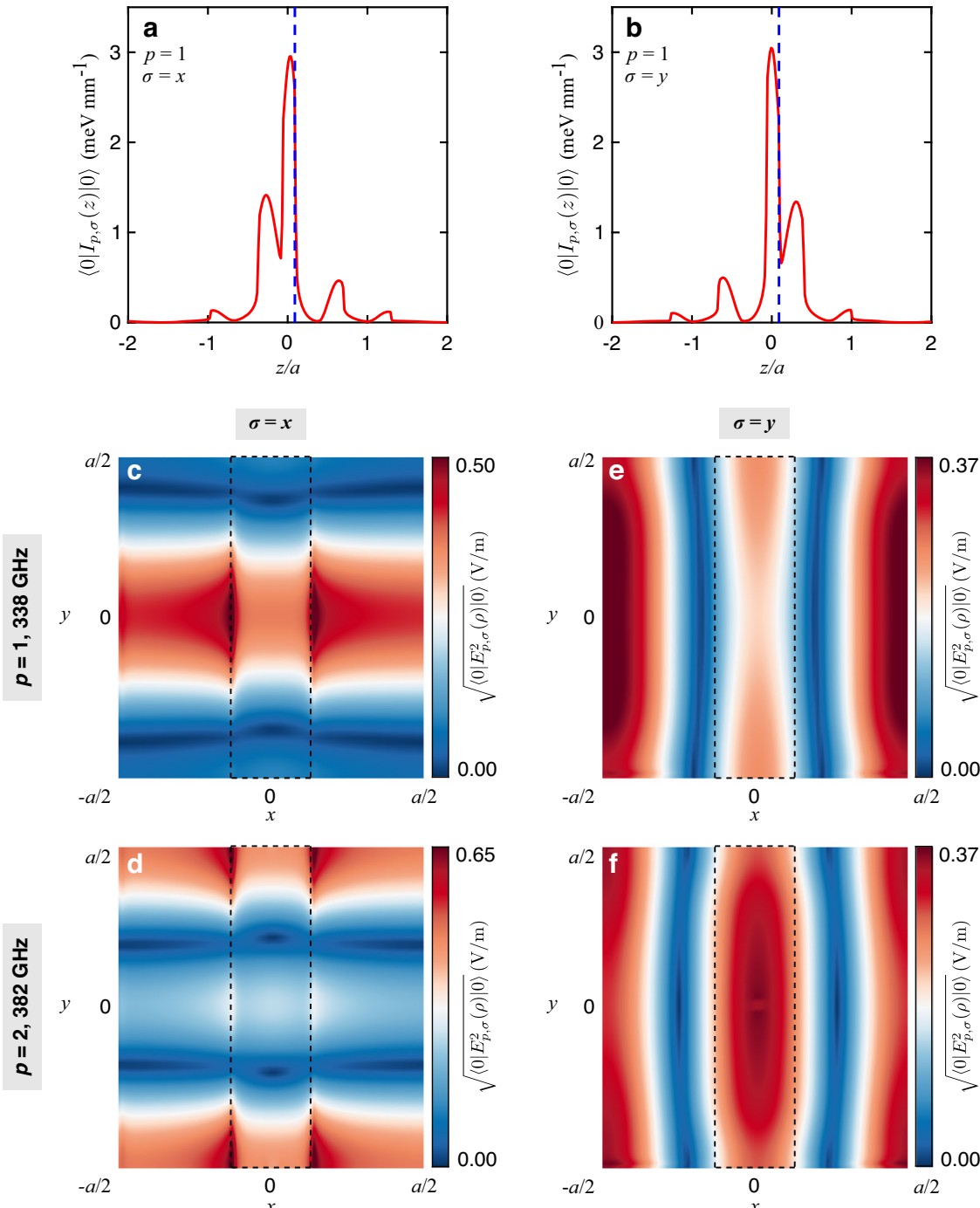

**Fig. 2 | Polarization-dependent mode profiles.** Expectation value $\langle 0|I_{p,\sigma}(z)|0\rangle$ of the electric energy density along $z$ in the vacuum state $|0\rangle$ for **a** the $\sigma = x$ and **b** $\sigma = y$ polarizations and the mode $p = 1$ (see Supplementary Section 1). The blue dashed line denotes the 2DEG location. The standard deviation $\sqrt{\langle 0|E_{p,\sigma}^2(\boldsymbol{\rho})|0\rangle}$ of the in-plane electric field in the vacuum state (see Supplementary Section 1) at $z = z_{2\text{DEG}}$ for the $\sigma = x$ (**c**, **d**) and $\sigma = y$ (**e**, **f**) polarizations. The upper and lower panels correspond to the mode $p = 1$ ($\omega_1/2\pi = 338$ GHz) and $p = 2$ ($\omega_2/2\pi = 382$ GHz), respectively. The dashed rectangle at the center illustrates the silicon rod that is right above the 2DEG layer.

at each position $\boldsymbol{\rho}$ are extra degrees of freedom that are introduced in the effective Hamiltonian $\hat{H}_{\text{CR}}$ because they happen to be the modes coupled to the electromagnetic field (see below).

The next term in the Hamiltonian,

$$
\begin{aligned}
\hat{H}_{\text{int}} = i\hbar \sum_{p,\sigma} \int \frac{d\boldsymbol{\rho}}{a} g_{p,\sigma,x}(\boldsymbol{\rho}) \left[ \hat{b}^\dagger(\boldsymbol{\rho}) - \hat{b}(\boldsymbol{\rho}) \right] \left( \hat{a}_{p,\sigma} + \hat{a}_{p,\sigma}^\dagger \right) \\
+ \hbar \sum_{p,\sigma} \int \frac{d\boldsymbol{\rho}}{a} g_{p,\sigma,y}(\boldsymbol{\rho}) \left[ \hat{b}^\dagger(\boldsymbol{\rho}) + \hat{b}(\boldsymbol{\rho}) \right] \left( \hat{a}_{p,\sigma} + \hat{a}_{p,\sigma}^\dagger \right),
\end{aligned}
\tag{2}
$$

is the linear coupling between the CR and the electromagnetic field, whose strength is $g_{p,\sigma,j}(\boldsymbol{\rho}) = E_{p,\sigma,j}(\boldsymbol{\rho}, z_{2\text{DEG}}) \sqrt{e^2 \omega_c n_e/(4\varepsilon_0 m_{\text{eff}} \omega_p a)}$. Here $j = x, y$ denotes the different components of the (real, dimensionless) electric field mode functions $E_{p,\sigma,j}(\boldsymbol{\rho}, z_{2\text{DEG}})$ at the in-plane position $\boldsymbol{\rho}$ and vertical location of the quantum well, $m_{\text{eff}}$ is the electron mass in GaAs, $\varepsilon_0$ is the vacuum permittivity, and $n_e$ is the total electron density. The coupling of the CR excitations to each cavity mode ($p, \sigma$) is weighted by the electric field mode functions since the electric field in the 3D-PCC is not uniform in the $xy$ plane, in contrast to 1D-PCCs[11,12].

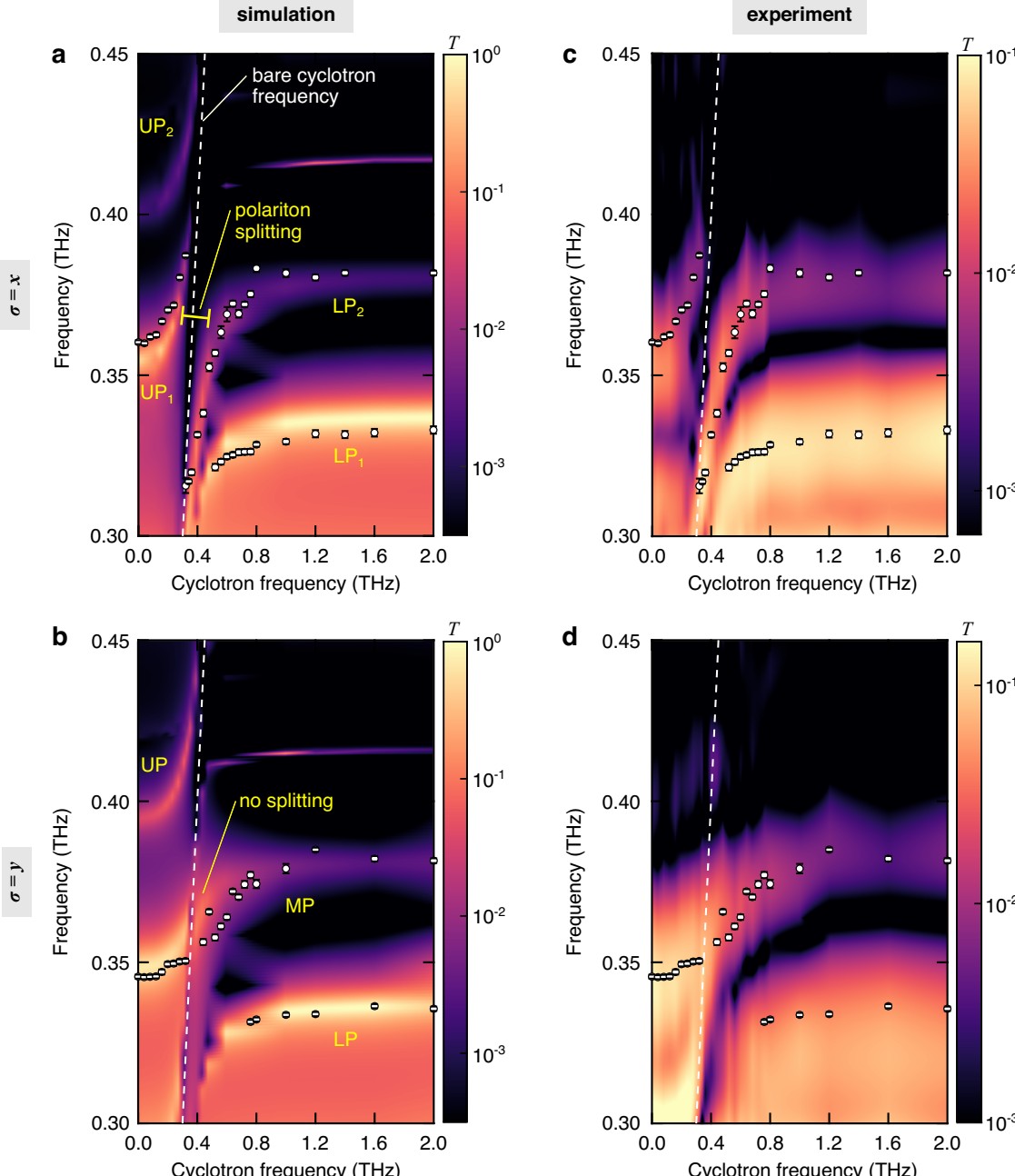

**Fig. 3 | Polarization-dependent mixing of photonic modes.** Transmittance ($T$) spectra as a function of cyclotron frequency, $\omega_c/(2\pi)$, obtained from **a**, **b** numerical simulations and **c**, **d** experiments. The top (bottom) panels are for $\sigma = x$ ($\sigma = y$) polarizations. The white dots denote the peak frequencies extracted from experimental data using longer time-domain traces (Supplementary Section 3 and Supplementary Fig. 15). The error bars of the white dots are discussed in Supplementary Section 3. The white dashed line shows the bare cyclotron frequency, $\omega_c/(2\pi)$ (Supplementary Fig. 8). The LP–UP splitting is marked.

Nevertheless, since the electric field mode functions are quasi-uniform over a cyclotron orbit, i.e., $a \ll l_c$, the dipole approximation can still be used to calculate the coupling matrix elements.

The last term in the Hamiltonian, the so-called $A^2$ term,

$$\hat{H}_{A^2} = \sum_{p,p'} \sum_{\sigma,\sigma'} \hbar D_{p,p';\sigma,\sigma'} \left( \hat{a}_{p,\sigma} + \hat{a}_{p,\sigma}^\dagger \right) \left( \hat{a}_{p',\sigma'} + \hat{a}_{p',\sigma'}^\dagger \right), \quad (3)$$

is responsible for direct coupling between the cavity modes. This term scales with $D_{p,p';\sigma,\sigma'} = \sum_j \int (d\boldsymbol{\rho}/a^2) g_{p,\sigma,j}(\boldsymbol{\rho}) g_{p',\sigma',j}(\boldsymbol{\rho})/\omega_c \propto \sum_j \int d\boldsymbol{\rho} E_{p,\sigma,j}(\boldsymbol{\rho}, z_{2\text{DEG}}) E_{p',\sigma',j}(\boldsymbol{\rho}, z_{2\text{DEG}})$, and thus, involves an overlap integral between the in-plane spatial profiles of the modes. It should be emphasized that the cavity modes are normalized over the entire

volume of the 3D-PCC, i.e., $\int d\boldsymbol{\rho} dz \, \varepsilon(\boldsymbol{\rho}, z) \mathbf{E}_{p,\sigma}(\boldsymbol{\rho}, z) \cdot \mathbf{E}_{p',\sigma'}(\boldsymbol{\rho}, z) = a^3 \delta_{p,p'} \delta_{\sigma,\sigma'}$, with $\varepsilon(\boldsymbol{\rho}, z)$ the inhomogeneous dielectric profile. Thus, the cavity modes are orthogonal if the entire volume of the 3D-PCC is considered. Nevertheless, the in-plane overlap of the cavity mode profiles, which governs the mixing of the cavity modes mediated by the CR, is a priori finite, as was recently reported in THz metamaterial resonators[24–26]. When the in-plane overlap is small, each cavity mode couples to the CR independently, and no matter-mediated intermode coupling exists. Each cavity mode leads to two polariton branches, resulting in a total of 2$N$ branches[24,26–29] (see Fig. 4a for $N = 2$). A splitting between the LPs and UPs along the bare CR frequency appears. When the spatial overlap of the cavity modes increases, the cavity modes can couple to each other through the matter, and the

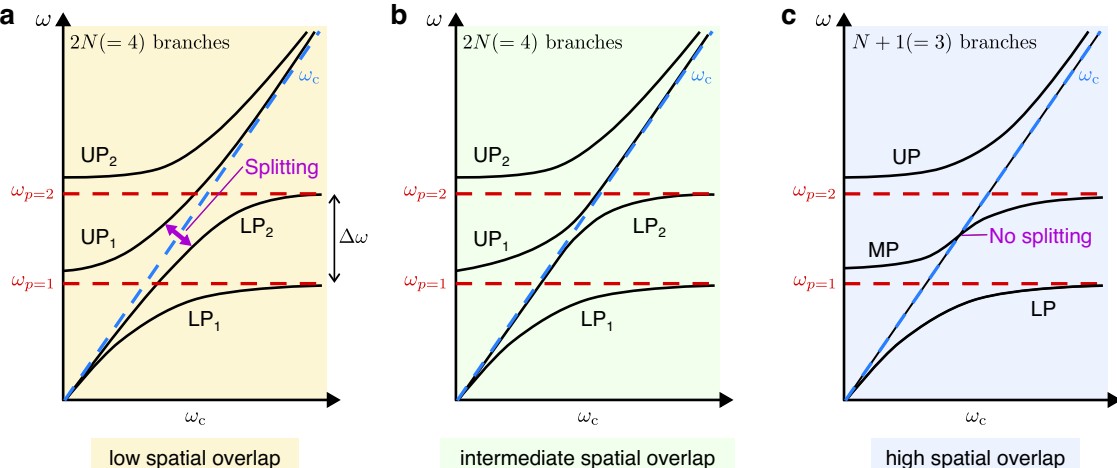

**Fig. 4 | Different scenarios of multimode light−matter coupling. a−c** Sketch of the typical polariton branches (black solid lines) as a function of the magnetic field for two cavity modes ($N = 2$, red dashed lines). The CR is depicted as a blue dashed line, $\omega_c$. **a** When the spatial overlap between the cavity mode profiles is low, the cavity modes coupled independently to the CR, resulting in $2N$ polariton branches ("decoupled scenario"). A splitting occurs between the LPs and UPs. **b** As the spatial overlap increases, the cavity modes start mixing with each other via the CR, and the splitting gradually decreases. **c** When the cavity mode profiles exhibit high spatial overlap, the splitting vanishes, leading to $N + 1$ polariton branches and an additional dark state along the $\omega_c$ line ("coupled scenario"). In this case, UP$_1$ and LP$_2$ merge to form an MP, which is a mixture of the two cavity modes close to the inflection point. UP upper polariton, LP lower polariton, MP middle polariton.

gap between the LPs and UPs decreases (see Fig. 4b for $N = 2$). When the different cavity modes within the active region highly overlap, these modes efficiently hybridize through the matter. The LP−UP splitting vanishes, producing $N + 1$ polariton branches with an S-shaped middle polariton (MP) and an additional dark state along the CR line (see Fig. 4c for $N = 2$). Furthermore, when the light−matter coupling strength is comparable or larger than the frequency difference between the cavity modes, the system enters the super-strong coupling (SSC) regime[30−36], and the mixing between different cavity modes increases, especially close to the inflection point of the S-shaped MP (Fig. 4b).

To better understand how the two regimes with either mode coupling or mode decoupling emerge from the microscopic description, we introduce new CR excitation operators following the spatial profiles of the cavity modes, $\hat{b}_{p,\sigma} = \int (d\boldsymbol{\rho}/a)(\widetilde{g}_{p,\sigma}(\boldsymbol{\rho})/\Omega_{p,\sigma})\hat{b}(\boldsymbol{\rho})$, with $\widetilde{g}_{p,\sigma}(\boldsymbol{\rho}) = g_{p,\sigma,y}(\boldsymbol{\rho}) - ig_{p,\sigma,x}(\boldsymbol{\rho})$ and the effective coupling strength $\Omega_{p,\sigma}$ defined by $\Omega_{p,\sigma}^2 = \int (d\boldsymbol{\rho}/a^2)|\widetilde{g}_{p,\sigma}(\boldsymbol{\rho})|^2$. By construction, the linear coupling term (2) written in terms of those operators takes the simple form

$$\hat{H}_{\text{int}} = \sum_{p,\sigma} \hbar\Omega_{p,\sigma}\left(\hat{b}_{p,\sigma} + \hat{b}_{p,\sigma}^\dagger\right)\left(\hat{a}_{p,\sigma} + \hat{a}_{p,\sigma}^\dagger\right). \quad (4)$$

The commutation relations between the new CR modes read $[\hat{b}_{p,\sigma}, \hat{b}_{p',\sigma'}^\dagger] = \xi_{p,p';\sigma,\sigma'}$, where

$$\xi_{p,p';\sigma,\sigma'} = \frac{1}{\Omega_{p,\sigma}\Omega_{p',\sigma'}}\int \frac{d\boldsymbol{\rho}}{a^2}\widetilde{g}_{p,\sigma}(\boldsymbol{\rho})\widetilde{g}_{p',\sigma'}^*(\boldsymbol{\rho}), \quad (0 \le |\xi_{p,p';\sigma,\sigma'}| \le 1) \quad (5)$$

is proportional to the in-plane spatial overlap of the different cavity modes, similarly to the off-diagonal contributions of the $A^2$ term. If the new CR modes $\hat{b}_{p,\sigma}$ were orthogonal, which is a priori not the case, one would have $\xi_{p,p';\sigma,\sigma'} = \delta_{p,p'}\delta_{\sigma,\sigma'}$. The deviation of the parameter $\xi_{p,p';\sigma,\sigma'}$ with respect to $\delta_{p,p'}\delta_{\sigma,\sigma'}$, therefore, is a measure of how much coupling between the cavity modes is mediated by the CR.

First, we find that $\xi_{p,p';x,y} < 0.36$ for all $p, p'$, indicating that the cavity modes with orthogonal polarizations are weakly coupled by the CR. Thus, the polarization index $\sigma$ remains quite a good quantum number to classify the polariton eigenmodes, as stated before. For a given polarization $\sigma$, $\xi_{p,p';\sigma,\sigma} = 1$ ($\xi_{p,p';j,j} = 0$) corresponds to

perfect overlap (no overlap) between the cavity modes $p$ and $p'$, in which case the cavity modes are coupled (decoupled) via the CR. For our 3D-PCC system, we find a rather small overlap between the spatial profiles of the $\sigma = x$ modes $p = 1$ and $p = 2$ ($\xi_{1,2;x,x} = 0.29$), suggesting that the intermode coupling is weak. Conversely, for $\sigma = y$ modes, the spatial profiles of cavity modes $p = 1$ and $p = 2$ highly overlap ($\xi_{1,2;y,y} = 0.91$), suggesting that they are strongly coupled through the CR.

When $\xi_{p,p';\sigma,\sigma'} = \delta_{p,p'}\delta_{\sigma,\sigma'}$, and neglecting the coupling between different polarizations induced by the $A^2$ term, one has $D_{p,p';\sigma,\sigma} \approx (\Omega_{p,\sigma}\Omega_{p',\sigma}/\omega_c)\xi_{p,p';\sigma,\sigma} = (\Omega_{p,\sigma}^2/\omega_c)\delta_{p,p'}$, and the full Hamiltonian can then be simplified to a "decoupled" Hamiltonian of the form $\hat{H} = \hat{H}_{\text{cav}} + \hat{H}_{\text{CR}} + \hat{H}_{\text{int}} + \hat{H}_{A^2}$, with $\hat{H}_{\text{cav}} = \sum_{p,\sigma}\hbar\omega_p\hat{a}_{p,\sigma}^\dagger\hat{a}_{p,\sigma}$, $\hat{H}_{\text{CR}} = \hbar\omega_c\sum_{p,\sigma}\hat{b}_{p,\sigma}^\dagger\hat{b}_{p,\sigma}$, $\hat{H}_{\text{int}}$ given by Eq. (4), and

$$\hat{H}_{A^2} = \sum_{\sigma,p}\frac{\hbar\Omega_{p,\sigma}^2}{\omega_c}\left(\hat{a}_{p,\sigma} + \hat{a}_{p,\sigma}^\dagger\right)^2. \quad (6)$$

In contrast to the full Hamiltonian, such a decoupled Hamiltonian can be diagonalized in each subspace $(p, \sigma)$ independently[28].

We computed transmittance spectra by extending the input−output model of ref. 37 in a simple planar geometry to a multimode PCC cavity (Supplementary Section 1). Dissipation is included through the (Markovian) coupling of cavity modes and CR excitations to phenomenological bosonic reservoirs, with associated quality factors of the bare cavity modes computed by FDTD, and of the intrinsic CR decay rate, respectively (Supplementary Section 1). The input−output model allows to select the polarization ($x, y$) of the input THz field to probe the polariton eigenmodes for both $\sigma = x$ and $\sigma = y$. The calculated spectra using the full Hamiltonian (Fig. 5a, b) are consistent with the simulations and experimental results (Fig. 3), except for the transmittance background at low frequencies due to the lower edge of the photonic band gap that is not included in the model. In particular, the model replicates the existence of an LP−UP splitting close to the $\omega_c$ line and the emergence of an S-shaped MP that adiabatically transforms from the UP of mode 1 to the LP of mode 2 for $\sigma = x$ and $\sigma = y$ modes, respectively. Note that since the input field in the model is assumed to be independent of the cavity mode it couples to (Supplementary Section 1), the transmittance at high frequencies is

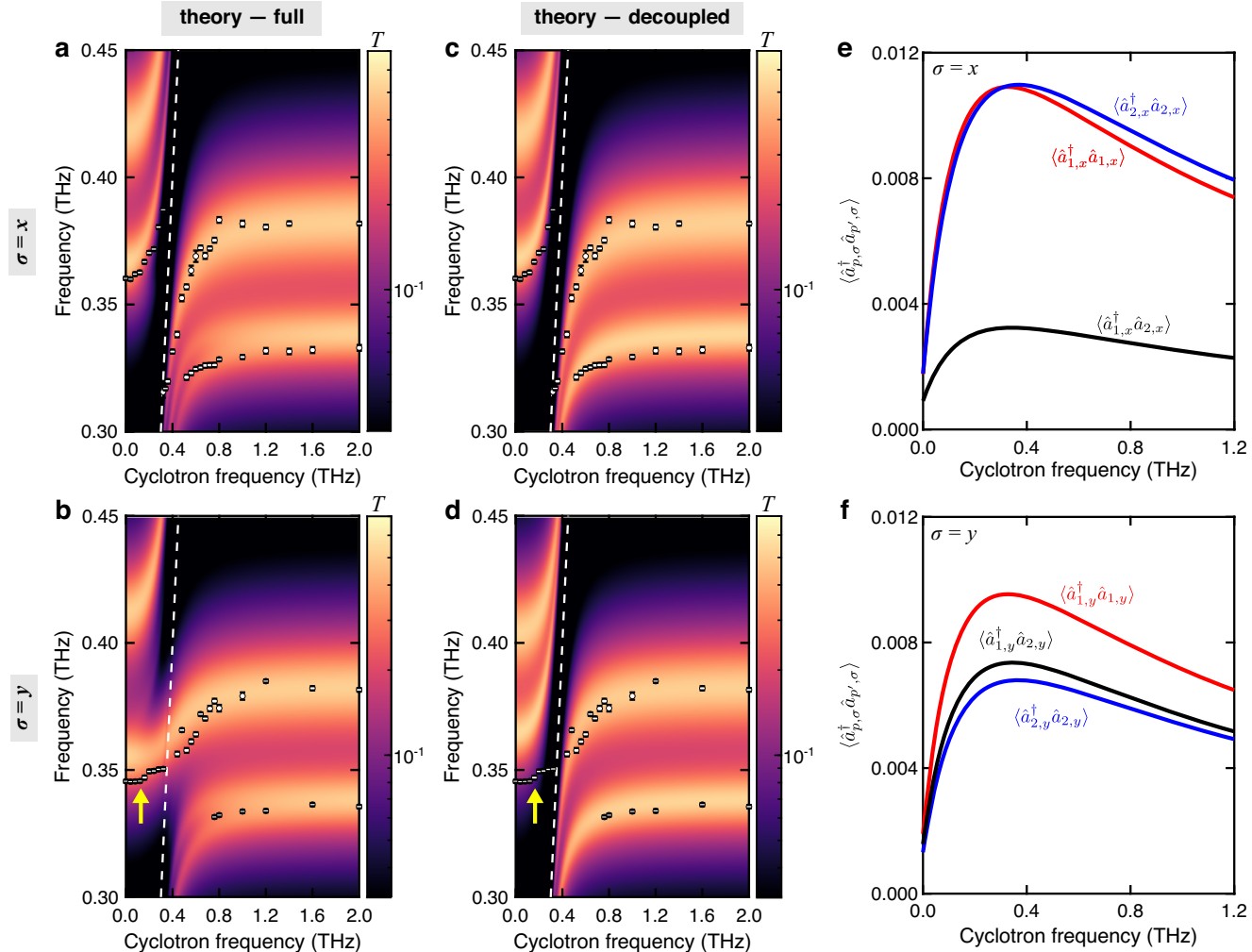

**Fig. 5 | Extended Hopfield model for multimode light–matter coupling in the 3D-PCC.** Transmittance ($T$) spectra as a function of cyclotron frequency, $\omega_c/(2\pi)$, obtained from calculations using **a**, **b** the full Hamiltonian and **c**, **d** the decoupled Hamiltonian, respectively (Supplementary Section 1). The top (bottom) panels are for $\sigma = x$ ($\sigma = y$) modes. The white dots denote the peak frequencies extracted from experimental data using longer time-domain traces (Supplementary Section 3 and Supplementary Fig. 15). The error bars of the white dots are discussed in Supplementary Section 3. The white dashed line represents $\omega_c/(2\pi)$. The yellow arrows in (**b**, **d**) mark the UP that significantly deviates from the experimental data points in the calculation using the decoupled Hamiltonian. Calculated ground-state (virtual) correlations $\langle \hat{a}^{\dagger}_{p,\sigma} \hat{a}_{p',\sigma} \rangle$ using the full Hamiltonian for **e** $\sigma = x$ and **f** $\sigma = y$ modes. The intermode ground-state correlations between the photonic modes $p = 1$ and $p = 2$ are significant for the $\sigma = y$ modes.

overestimated as compared to the experimental data and numerical simulations. Since $\xi_{1,2;x,x}$ ($\sigma = x$) is small, the decoupled model provides a faithful description of the system, and thus the calculated spectra (Fig. 5c) are almost the same as that using the full Hamiltonian (Fig. 5a). For $\sigma = y$, however, the decoupled model is inaccurate as it fails to predict the emergence of the S-shaped MP (Fig. 5d). This is due to the large overlap $\xi_{1,2;y,y} = 0.91$, which means that the CR operators $\hat{b}_{1,y}$ and $\hat{b}^{\dagger}_{1,y}$ are not orthogonal, i.e., $[\hat{b}_{1,y}, \hat{b}^{\dagger}_{2,y}] \neq 0$.

In order to assess in which regime of light–matter coupling our system operates, we extracted the coupling strengths for each cavity mode from the theoretical model and evaluated the relevant figures of merit. The USC regime, which has attracted widespread interest due to the compelling prospect of manipulating basic material properties by engineering the vacuum electromagnetic field surrounding the material inside a cavity[12,38,39], is achieved when the effective coupling strength for each cavity mode $\Omega_{p,\sigma}$ reaches about 10% of the bare mode frequency at resonance[18,19]. Here we find $\Omega_{1,x}/\omega_1 \approx 0.21$ and $\Omega_{1,y}/\omega_1 \approx 0.2$ at $B = 0.81$ T, and $\Omega_{2,x}/\omega_2 \approx 0.21$ and $\Omega_{2,y}/\omega_2 \approx 0.17$ at $B = 0.92$ T, respectively. This shows that our system operates in the USC regime for both cavity modes and polarization $\sigma$.

It is interesting to investigate unique quantum features that can emerge in our multimode system. Since the ground state in the USC regime is known to be a dressed ground state that has a finite photon population, $\langle n_{p,\sigma} \rangle = \langle \hat{a}^{\dagger}_{p,\sigma} \hat{a}_{p,\sigma} \rangle$[37,40], an intriguing question is whether the virtual photons (cavity vacuum fields) in different modes can mix through simultaneous USC with the matter, resulting in finite intermode correlations in the ground state $\langle \hat{a}^{\dagger}_{p,\sigma} \hat{a}_{p',\sigma} \rangle > 0$. We computed the photon ground-state correlations $\langle \hat{a}^{\dagger}_{p,\sigma} \hat{a}_{p',\sigma} \rangle$ for both $\sigma = x, y$ by inverting the Bogoliubov transformation that diagonalizes the full (quadratic) Hamiltonian. The photon number for each mode $p$ and polarization $\sigma$ in the ground state (without external driving) is shown in Fig. 5e. While $\langle \hat{a}^{\dagger}_{p,\sigma} \hat{a}_{p,\sigma} \rangle$ reaches its maximum at resonance between the CR and the cavity mode ($p, \sigma$), the intermode correlation $\langle \hat{a}^{\dagger}_{1,\sigma} \hat{a}_{2,\sigma} \rangle$ is peaked in between the two resonances. For $\sigma = x$, we find that $\langle \hat{a}^{\dagger}_{1,x} \hat{a}_{2,x} \rangle \ll \langle \hat{a}^{\dagger}_{p,x} \hat{a}_{p,x} \rangle$. However, for $\sigma = y$, $\langle \hat{a}^{\dagger}_{1,y} \hat{a}_{2,y} \rangle$ is comparable to $\langle \hat{a}^{\dagger}_{p,y} \hat{a}_{p,y} \rangle$ (Fig. 5f), which suggests that the mixing of virtual photons in different photonic modes is also directly related to the spatial overlap of the cavity profiles. Those

intermode vacuum correlations should be accessible in our system from, e.g., first-order equal-time correlation measurements.

Based on the scaling of the intermode ground-state correlations with the overlap between the cavity modes and the mode frequencies (see Supplementary Fig. 1b), we extend the standard figure of merit for single-mode USC to the multimode case by introducing the parameter

$$\eta_{pp',\sigma} \equiv \sqrt{\frac{\int (d\boldsymbol{\rho}/a^2) \widetilde{g}_{p,\sigma}(\boldsymbol{\rho}) \widetilde{g}^*_{p',\sigma}(\boldsymbol{\rho})}{\omega_c(\omega_p + \omega_{p'})/2}}, \qquad (7)$$

which does not depend on the magnetic field and whose diagonal elements ($p = p'$) coincide with the standard figure of merit. While intermode (off-diagonal) coupling for $\sigma = x$ is smaller than the diagonal coupling ($\eta_{12,x} = 0.11$ as compared to $\eta_{11,x} \approx \eta_{22,x} \approx 0.21$), we find that intermode coupling for $\sigma = y$ is of the same magnitude ($\eta_{12,y} = 0.17$ as compared to $\eta_{11,y} \approx 0.19$ and $\eta_{22,y} \approx 0.16$), consistently with the intermode ground-state correlations.

On the other hand, the SSC regime is achieved when the coupling strength becomes comparable to the frequency difference between the uncoupled modes, thereby signaling the breakdown of the single-mode approximation[30–36]. In this regime, the hybridization of cavity mode profiles[30] and complex multimode dynamics[32,41] have been discussed. However, our results highlight that such a criterion is not sufficient to characterize the SSC regime because the mixing between cavity modes is also influenced by the spatial profiles of the cavity modes. For example, two independent sets of LPs and UPs are observed for $\sigma = x$ as a result of the weak overlap of the cavity modes. Thus, the single-mode approximation still holds in this case even though the standard criterion is satisfied. As explained above, this "mode decoupling" situation would generally occur in all systems where quantum emitters fill the entire cavity mode volume[24,26–29]. To circumvent this issue, we extend the standard figure of merit for the SSC regime to take into account the spatial overlap between the cavity modes,

$$\Lambda_\sigma \equiv \sqrt{\frac{\int (d\boldsymbol{\rho}/a^2) \widetilde{g}_{1,\sigma}(\boldsymbol{\rho}) \widetilde{g}^*_{2,\sigma}(\boldsymbol{\rho})}{\omega_c(\omega_2 - \omega_1)}}. \qquad (8)$$

This parameter quantifies the strength of the intermode coupling with respect to the frequency difference $\omega_2 - \omega_1$, including the effect of the cavity mode overlap. This parameter is $B$-independent as $\omega_c$ is included in the denominator, which balances the $\omega_c$ term in the expression for $\widetilde{g}_{1,\sigma} \widetilde{g}^*_{2,\sigma}$. We find that $\Lambda_x \approx 0.33$ and $\Lambda_y \approx 0.49$, which indicates that the polarization $\sigma = y$ is further in the SSC regime than the polarization $\sigma = x$. Deep into the SSC regime ($\Lambda_\sigma \sim 1$), the S-shaped MP would become a hybrid mode equally composed of the two cavity modes. We show in Supplementary Section 1 that the SSC figure of merit governs the weight of the MP onto the different cavity modes (Supplementary Fig. 1c), which in turn governs the intermode correlations in polaritonic excited states.

## Discussion

We reported ultrastrong and SSC in a THz 3D-PCC coupled to a Landau-quantized 2DEG in GaAs. The in-plane reciprocal lattice vector, $G$, and the spatial profiles of the cavity modes of the 3D-PCC affect the coupling between different cavity modes mediated by the matter. In contrast to typical planar cavities with mirror symmetry in the stacking direction, our cavity incorporates a symmetry that combines 4-fold rotational symmetry around the $z$-axis and mirror symmetry in the $z$-direction. This unique symmetry enables distinct degrees of hybridization between cavity modes that can be selectively observed by rotating the polarization of the incident light while preserving USC between the CR and each cavity mode. Interestingly, a similar phenomenon was recently observed using a different approach in a THz

metasurface array[26]. Despite the inherent orthogonality of the cavity mode profiles over the entire unit cell of the resonator, the spatial overlap remains finite when only a subregion of the resonator is considered. To exploit this, Mornhinweg et al. patterned the quantum well to confine the active region to the central gap area, where the mode profiles highly overlap. These complementary works highlight the crucial role of spatial profiles in multimode coupling and emphasize the potential of engineered mode profiles and spatial overlap optimization in advancing our understanding of light–matter interaction in extreme regimes.

Our results emphasize that the coupling strength of a multimode system cannot be accurately extracted from the vacuum Rabi splitting in the spectrum due to the formation of MPs when the cavity mode profiles highly overlap. Our experimental data showed good agreement with simulations and the calculations based on a microscopic model. We discussed the possible existence of quantum ground-state correlations between two photonic modes in the multimode USC regime that were derived from the theoretical model. We introduced relevant figures of merit to characterize the light–matter coupling regimes in multimode systems.

Although our system is linear in the sense that it is well described by a quadratic Hamiltonian, and thus does not feature photon-photon many-body interactions, the latter could be achieved by, e.g., replacing GaAs with a non-parabolic semiconductor or graphene, which would lead to a nonlinear CR. Introducing such nonlinearities in our system that operates in the USC and the SSC regimes simultaneously thus opens up interesting perspectives to explore both multimode vacuum-induced effects[25,42–45] and driven-dissipative dynamics in the many-body regime of quantum optics[31,32,34,46].

The 3D-PCC design is highly versatile and can be tailored to achieve photonic modes with smaller mode volumes and higher quality factors, for example, using a point defect[47]. Our approach can be utilized to investigate the coupling between the topological edge states of a 2DEG and the topological edge states of a chiral woodpile structure[48–52]. Moreover, the photonic modes with well-defined in-plane wave vectors in the 3D-PCC satisfy the condition to circumvent the no-go theorem for the Dicke SRPT[1–3,8]. While the photon wave-vector $\sim 2\pi/a$ is limited by the lattice parameter $a$ of the 3D-PCC and remains relatively small compared to the electron momentum $\sim 1/l_c$, the possibility of fulfilling the criterion for 2D photon condensation in this regime has been recently discussed[2]. Future work can focus on the predicted renormalization of matter's kinetic energy and effective mass at finite in-plane wave vectors near the photonic band edge[16].

## Methods

### Preparation of the multiple quantum well sample

A wafer containing multiple GaAs quantum wells (MQW) was grown by the Purdue molecular beam epitaxy system. This structure had ten 30-nm-thick GaAs MQW separated by 160-nm $Al_{0.24}Ga_{0.76}As$ barriers. Silicon dopants were placed 80 nm away from the GaAs MQW. An electron density per well of $3.08 \times 10^{11}\, cm^{-2}$ and a mobility of $2.36 \times 10^7\, cm^2/Vs$ were extracted from Hall transport measurements at 300 mK in the dark. The total electron density of the MQW was $3.08 \times 10^{12}\, cm^{-2}$.

### Fabrication of the THz woodpile cavity

A 100-μm-thick silicon wafer was coated by lift-off resist and photoresist layers through spin coating. The layers were patterned via photolithography. The rod width and periodicity on the mask were designed as 87 μm and 333 μm, respectively. An $\sim$1.2-μm-thick $Al_2O_3$ layer was deposited on the coated silicon wafer by an e-beam evaporator. The lift-off process was completed by removing the resist with Remover PG; leaving $Al_2O_3$ on the patterned area. Afterward, the silicon wafer was etched by deep reactive ion etching using a Bosch process. During this process, the silicon wafer was cut into multiple

smaller pieces, and the gaps between silicon rods and holes at the corners were also created in each piece of the smaller wafer. The remaining $Al_2O_3$ layer was removed by immersing the sample in a heated 1:3 solution of phosphoric acid and sulfuric acid. A custom-made sample holder was used for stacking the silicon wafers to form a woodpile cavity. The orientation of the silicon rods was fixed by the screws that passed through the holes at the corners. The woodpile cavity can be conveniently disassembled and reassembled.

### THz time-domain magnetospectroscopy measurements

Transmittance spectra for samples were measured by a home-built THz time-domain magnetospectroscopy setup[11,12,53–61]. The near-infrared (775 nm) output beam of a Ti:sapphire regenerative amplifier (1 kHz, 200 fs, Clark-MXR, Inc., CPA-2001) was split into pump and probe beams. The pump beam was used to generate linearly polarized THz radiation in a ZnTe crystal through optical rectification. The generated THz beam was focused onto the sample that was mounted on the sample holder of a commercial 10-T superconducting magnet cryostat (Oxford Instruments, Inc., Spectromag). The magnetic field was factory-calibrated. The sample was cooled down with helium gas inside the variable temperature insert of the cryostat, with a temperature range between 1.4 and 300 K. All measurements with the MQW sample were collected at 4 K. The sample temperature was controlled by a PID temperature controller. The direction of the static magnetic field was normal to the sample surface. The measurements with a bare cavity were conducted using a commercial THz-TDS (Toptica, TeraFlash Pro) at room temperature.

The time-domain waveform of the transmitted THz radiation, $E_{sample}(t)$, was measured with another ZnTe crystal via electro-optic sampling with controlled time delays. A reference signal, $E_{reference}(t)$, was measured by repeating the measurements in the absence of a sample. $E_{sample}(t)$ and $E_{reference}(t)$ were Fourier-transformed into complex-valued frequency-domain spectra, $\tilde{E}_{sample}(\omega)$ and $\tilde{E}_{reference}(\omega)$, respectively. Transmittance, $T$, is equal to $|\tilde{E}_{sample}(\omega)|^2/|\tilde{E}_{reference}(\omega)|^2$. The extraction of peak frequencies is discussed in Supplementary Section 3 (Supplementary Figs. 13 and 15).

## Data availability

The raw data generated in this study is available at the Rice Research Repository (R-3) under https://hdl.handle.net/1911/118272.

## Code availability

The details of the simulations are described in Supplementary Information. Additional information related to this paper is available from the corresponding author upon request.

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

## Acknowledgements

The authors thank Motoaki Bamba, Kaden Hazzard, Tal Schwartz, Thibault Chervy, and Cyriaque Genet for useful discussions. J.K. acknowledges support from the U.S. Army Research Office (through Award No. W911NF2110157), the Gordon and Betty Moore Foundation (through Grant No. 11520), the W. M. Keck Foundation (through Award No. 995764), and the Robert A. Welch Foundation (through Grant No. C-1509). This work was done in part using resources of the Research Support Shop and Shared Equipment Authority at Rice University, as well as computational resources of the Centre de calcul de l'université de Strasbourg (CCUS).

## Author contributions

F.T., D.H., and J.K. conceptualized the project. F.T. designed the cavity device. A.M. fabricated the cavity device. F.T., A.B., and H.X. performed the THz measurements. F.T. analyzed the experimental data. D.H. derived the microscopic model and performed MEEP simulations and calculations. S.L., G.C.G., and M.J.M. grew the 2DEG sample by the molecular beam epitaxy system. F.T., S.S., and A.A. conducted COMSOL simulations. A.B., D.H., and J.K. supervised the project. F.T., D.H., and J.K. wrote the manuscript, with inputs from all authors.

## Competing interests

The authors declare no competing interests.
