## [Transparent Peer Review file · Nature Communications]

Multimode ultrastrong coupling in three-dimensional photonic-crystal cavities

Corresponding Author: Professor Junichiro Kono

Version 0:

Reviewer comments:

Reviewer #1

(Remarks to the Author)

The present manuscript by F. Tay reports on a novel approach for an ultrastrongly light-matter coupled structure based on Landau-quantized electrons in conventional semiconductor quantum wells and, in the given context, a highly innovative 3D photonic crystal cavity. They achieve coupling of several optical modes and observe anti-crossing behaviour depending on the polarization state. The data are very clear and of outstanding quality. The authors analyze their results by a state of the art theoretical approach which includes multiple modes, fractional mode overlap factors, as well as anti-resonant couplings.

This paper features several noteworthy highlights and firsts:

Flexibility. Ultrastrong and deep-strong coupling have been explored in various systems in the THz range, most of them based on metallic planar resonator structures. Only recently, the Kono group developed a Bragg-type high-Q resonator which addressed the problem of otherwise large linewidths, yet to some degree at the expense of flexibility compared to metal-type resonators. Here, they have opened the door to 3D structuring of resonator modes for high-Q THz cavities that can furthermore support polarization-dependent responses.

Multi-mode design. The PCC designed here, in combination with potentially large coupling strengths, enables designing multi-mode light-matter coupling with many participating optical and electronic modes across a significant spectral bandwidth, and with variable overlap factors (Fig. 4). In the future, the placement of the 2DEG relative to the modes, or even a combination of PCC and planar metal cavities, could further expand the reach of this approach.

Large bandwidth and cross-polarization coupling. What the authors currently find to be insignificant in the current implementation, namely the coupling of orthogonal cavity modes by the gyrotropic nature of the cyclotron resonance, would even allow for mixing of several polarization modes of the cavity, tunable by an external parameter - the magnetic bias field.

I thus clearly recommend the manuscript for publication in Nature Communications.

However, the authors should consider the following points:

- I find it unintuitive to plot the data in Figs. 3-5 as a function of the magnetic field which is by itself meaningless. The authors should consider switching to cyclotron frequency as in their previous papers.
- The authors should consider including the paper by Mornhinweg et al. demonstrating multi-mode coupling (Nature Comm. 15, 1847 (2024)).

(Remarks on code availability)

Reviewer #2

(Remarks to the Author)

I thank the authors for their willing efforts to improve and reinforce the manuscript in response to my first report. I am (almost) happy with the revised manuscript and I am ready to recommend publication once the authors have taken into account my second round of (relatively minor) remarks listed below.

Concerning the importance and the potential impact of the results, I keep my doubts that the results are worth publication on Nature Photonics, but I am quite convinced that they are at the level of Nature Comms. Of course eventual publication is the editor's decision, but I am personally well in favor of publication on Nature Comms.

---- remarks ----

-bottom of pag.4: the authors say that the energy density 'is maximum at the vertical location of the silicon rod...'. Looking at fig.2a,b I have the different impression that the energy is maximum at the center of the spacing layer (the maximum at $z=0$ is higher than the adjacent one). The authors should check and in case reformulate the text to avoid inconsistencies.

-bottom of pag.5: the authors speak (a couple of times) of 'Si log'... I suspect that they actually meant 'Si rod'. Please check.

-on pag.9, the authors should mention that the difference in the number of polariton branches in the two panels of fig.4 is compensated by an additional dark state at the cyclotron frequency. This would also help the readers to understand that the splitting in fig.4(a) originates from the anticrossing of the S-shaped band of fig.4(b) with a state located along the cyclotron line which start mixing with the other modes in going from (b) to (a).

-along similar lines, I might anticipate that adding a third panel to fig.4 showing an intermediate situation could clarify the presentation. As it does not require much extra work, the authors should take this idea in serious consideration.

-around pag.12: I wonder why the authors have focussed their attention on the ground-state correlations of the form $\langle a^\dagger + a \rangle$ and have not mentioned the anomalous ones $\langle a \rangle$. Could there be any interesting new physics in these latter?

-on pag.12, below eq.5: the authors should mention again the values of η for the diagonal coupling together with the ones for the non-diagonal coupling, so that readers can make themselves the comparison of the two cases.

-on pag.13: I do not understand why the parameter Λ in eq.(6) is said to be independent of ω_c . What factor compensates the ω_c in the denominator?

-a couple of lines later, I do not understand why the MP would become a pure-matter mode if it is composed of the two cavity modes. Please check and clarify.

-The authors should mention in a more explicit way that a similar physics has been very recently investigated in Ref.[25], even though on a completely different physical setup. As the two works were submitted (to the best of my understanding) around the same time, this fact does not reduce the importance of the present work nor raises any concern about scientific priority. Still, readers may take benefit of being pointed to this other interesting work, possibly with some commenting sentences.

(Remarks on code availability)

Reviewer #3

(Remarks to the Author)

Review of manuscript NCOMMS-24-48838-T entitled "Multimode Ultrastrong Coupling in Three-Dimensional Photonic-Crystal Cavities" by Tay, Mojibpour, Sanders, Liang, Xu, Gardner, Baydin, Manfra, Alabastri, Hagenmüller, and Kono.

The manuscript presents a study of a 3D photonic crystal in the one-terahertz frequency range, with a planar defect containing a 2D electron gas that reveals so-called ultrastrong coupling with the confined fields. The study combines experiments, computer simulations, and theory, where the experiments are the central contribution.

I have a number of concerns on the paper (numbered as broad categories below), namely (1) on the paper as a whole, notably the broad impact; (2) on the sample; (3) on the experiments; (4) on the simulations.

1) Broad impact:

Let us discuss main features of the manuscript, using keywords from the title and abstract:

A) Multimode: I am not sure what is meant with this keyword, and why "multimode" is so remarkable when every real system is necessary multimode, for instance, always described by a dense set of states (and wave vectors).

B) Ultrastrong coupling: While "strong coupling" has been a topic of considerable interest in cavity QED - albeit accompanied by considerable debate on what is really meant by the notion - it is less clear how "ultrastrong coupling" is defined, and unfortunate that the manuscript does not really explain it.

C) Three-dimensional photonic crystal cavity: While I agree that the crystal structures top and bottom have a 3D structure, the central planar defect layer is structurally one-dimensional (1D), so I wonder if the confinement is truly 3D. I am ready to

be convinced that the 1D-confined EM waves experience lateral periodicity of both 3D structures, which would induce at most 2D periodicity in the physical properties. In the community, however, it is generally agreed that 3D cavities entail 3D confinement, which is not demonstrated here.

D) Terahertz frequencies: While I applaud challenging experiments as presented by this valiant team of authors, let us agree the range up to 1 THz may hold potential for interesting physics (in the present case the jury "is still out"), but the potential for large-volume device applications is exceedingly slim, since there are no practical devices operating at 1 THz and cryogenic temperatures. One may then argue that since EM is scale-invariant, studies like this one point lead way to optical applications. But this is also not convincing, given that the likelihood of realizing these phenomena at optical frequencies (>100 THz) seem far away.

2) Sample:

A) In most photonic crystal studies, considerable attention is paid to characterizing the detailed structure of newly fabricated crystal, and especially of the disorder. So it is unfortunate that this manuscript provides little detail on the realized sample; only a very low-resolution image is in the inset of Figure 1. It is relevant to provide close-up images of the sample both from the top and from the side, including cross-sections, to assess the sample quality. Given the fairly macroscopic character of the sample (layer thickness about 60 to 85 micron) it would be a great start to present optical microscopy (no need for electron microscopy.)

B) And complementarily, scattering studies would be very helpful to assess usual distributions that conveniently characterize disorder.

C) Finally the symmetry of the layers in the structure is surprising since there is no obvious symmetry. Naively, one would have expected the layers adjacent to the 1D planar defect to be of the same character (i.e. both woodpiles parallel or perpendicular.) If there is a compelling reason why the present choice was made, then it has unfortunately not become clear.

3) Experiments:

A) Setup: A description of the setup and experimental conditions is unfortunately missing. This could easily be described in the Supplementary. For instance, while I suspect that the experiments are being conducted at cryogenic temperatures. Similar about the magnetic field dependencies, which are impressive, but unfortunately no description is given how these are maintained (and calibrated).

B) Data: information is missing on essential experimental features that would help to assess the quality of the experiments. For instance, in both time and frequency domains questions arise: what is the noise floor and how does it depend on frequency, what is the spectral resolution? Etcetera.

C) Transmission: Early in the text, in Figure 1, one would hope to see the measured transmission of the fabricated structure (even better, for reproducibility: multiple structures?) Not only to assess ensuing steps in this optical study, but also to assess the sample quality. Deep in the supplementary, Figure S1 shows the measured transmission (aside: given that the scanning time-resolved measurement method is non-standard, it would be very helpful to see a number of legible raw measurements.) In Figure S1 it is first of all not clear why the simulations are fudged by 60% to match the experiments.

Secondly, it is very difficult to distinguish experimental features from simulations; for instance, the sharp peak near 0.48 THz (is it measured or computed?) Thirdly, why do the spectral shapes of apparently important peaks "1" and "2" differ markedly between simulations and experiments? What are the consequences for the claimed cavity-QED behaviour?

D) Figure S7: with all due respect, the data in all panels are completely unreadable. Why are so many traces plotted?

Please show a few representative traces and much enlarged. And while we're at it, please crop the abscissa if interesting features only occupy 1/3rd of the range.

About Figure S7 c): what do "a.u." mean (arbitrary units?) when the scale runs to 1.5×10^{-4} ? And, to appreciate the meaning of the various bursts, it would be helpful to plot amplitude on a semi-log scale (also to appreciate the noise floor) and cropped to discern underlying fringes?

4) Simulations & presentation:

A) In Figure 1, numerical simulations are shown with an impressive transmittance range, namely from 1 to less than 1×10^{-3} , or ~ 3.5 decades. Since in literature, one rarely sees such high precision, it would be helpful to describe what special measures were taken to arrive at this dynamic range?

B) In Figure 2, it is not clear how the (miniature) symbols are related to the shadings? Including, what are the error bars with all symbols?

In summary, since the manuscript is unfortunately deficient in many different aspects, publication in Nature Communications is not recommended.

(Remarks on code availability)

Version 1:

Reviewer comments:

Reviewer #1

(Remarks to the Author)

The authors have satisfactorily addressed the minor points I raised in my previous review.

I am moreover of the opinion that the extensive improvements they made in response to the other referees' questions have cleared smaller inconsistencies and strengthened the paper further, improving accessibility for non-experts significantly.

I recommend publication of the paper as is without any reservation.

Response Letter to Reviewers

Each comment is in blue, quoted in Italics, followed by our response. Major updates in the main text and Supplementary Information are highlighted in red fonts.

RESPONSE TO THE 1st REVIEWER:

The present manuscript by F. Tay reports on a novel approach for an ultrastrongly light-matter coupled structure based on Landau-quantized electrons in conventional semiconductor quantum wells and, in the given context, a highly innovative 3D photonic crystal cavity. They achieve coupling of several optical modes and observe anti-crossing behaviour depending on the polarization state. The data are very clear and of outstanding quality. The authors analyze their results by a state of the art theoretical approach which includes multiple modes, fractional mode overlap factors, as well as anti-resonant couplings.

This paper features several noteworthy highlights and firsts:

Flexibility. Ultrastrong and deep-strong coupling have been explored in various systems in the THz range, most of them based on metallic planar resonator structures. Only recently, the Kono group developed a Bragg-type high-Q resonator which addressed the problem of otherwise large linewidths, yet to some degree at the expense of flexibility compared to metal-type resonators. Here, they have opened the door to 3D structuring of resonator modes for high-Q THz cavities that can furthermore support polarization-dependent responses.

Multi-mode design. The PCC designed here, in combination with potentially large coupling strengths, enables designing multi-mode light-matter coupling with many participating optical and electronic modes across a significant spectral bandwidth, and with variable overlap factors (Fig. 4). In the future, the placement of the 2DEG relative to the modes, or even a combination of PCC and planar metal cavities, could further expand the reach of this approach.

Large bandwidth and cross-polarization coupling. What the authors currently find to be insignificant in the current implementation, namely the coupling of orthogonal cavity modes by the gyrotropic nature of the cyclotron resonance, would even allow for mixing of several polarization modes of the cavity, tunable by an external parameter - the magnetic bias field.

I thus clearly recommend the manuscript for publication in Nature Communications.

We sincerely thank the reviewer for the positive review and thoughtful evaluation of our manuscript. We greatly appreciate his/her summary of the key strengths and innovations in our work, as well as the insightful suggestions, which have helped us further enhance the manuscript.

However, the authors should consider the following points:

- I find it unintuitive to plot the data in Figs. 3-5 as a function of the magnetic field which is by itself meaningless. The authors should consider switching to cyclotron frequency as in their previous papers.

We thank the reviewer for the suggestion. We agree that using the cyclotron frequency as the horizontal axis provides a more meaningful representation. Accordingly, we have updated Fig. 3–5 to display the cyclotron frequency in place of the magnetic field.

- The authors should consider including the paper by Mornhinweg et al. demonstrating multi-mode coupling (Nature Comm. 15, 1847 (2024)).

We appreciate the reviewer's suggestion and have included the paper by Mornhinweg et al. in the manuscript.

RESPONSE TO THE 2ND REVIEWER:

I thank the authors for their willing efforts to improve and reinforce the manuscript in response to my first report. I am (almost) happy with the revised manuscript and I am ready to recommend publication once the authors have taken into account my second round of (relatively minor) remarks listed below.

Concerning the importance and the potential impact of the results, I keep my doubts that the results are worth publication on Nature Photonics, but I am quite convinced that they are at the level of Nature Comms. Of course eventual publication is the editor's decision, but I am personally well in favor of publication on Nature Comms.

We sincerely appreciate the reviewer's thoughtful and positive feedback, as well as his/her acknowledgement of our efforts to improve the manuscript. Once again, we thank the reviewer for valuable suggestions, which have provided us with valuable insights to further enhance the quality of this work.

---- remarks ----

-bottom of pag.4: the authors say that the energy density 'is maximum at the vertical location of the silicon rod...'. Looking at fig.2a,b I have the different impression that the energy is maximum at the center of the spacing layer (the maximum at $z=0$ is higher than the adjacent one). The authors should check and in case reformulate the text to avoid inconsistencies.

We thank the reviewer for this critical comment and apologize for any confusion caused by our earlier description. The description was inaccurate as the maximum of the energy density is indeed located at the center of the defect layer, as the reviewer pointed out. We intended to highlight that the spatial profile of energy density is asymmetric along the z direction. Specifically, the energy density is slightly higher near one of the defect layer boundaries adjacent to the silicon rod layer that is oriented perpendicular to the polarization of the incident THz light, compared to the opposite boundary. For instance, for $\sigma = x$

polarization, the energy density near the defect layer boundary at $z > 0$ is higher as the silicon rod layer adjacent to this boundary is oriented in the y direction. We have modified the paragraph to avoid any inconsistencies and confusion:

“Furthermore, the 3D-PCC features another symmetry: a combination of a 4-fold rotational symmetry with respect to the z -axis and an inversion symmetry with respect to the plane $z = 0$. This symmetry results in the degenerate frequency for the $\sigma = x$ and $\sigma = y$ modes in the spectrum [23] (see Supplementary Information Sec. 1). However, the lack of inversion symmetry leads to asymmetric cavity profiles along the z direction. Figure 2a,b display the expectation value of the electric energy density in the vacuum state $|0\rangle$, $\langle 0|I_{p,\sigma}(z)|0\rangle$, for the cavity modes $p = 1$ with $\sigma = x$ and $\sigma = y$. Therefore, despite having the same resonance frequency, the cavity modes excited by orthogonal polarizations exhibit different spatial profiles.”

-bottom of pag.5: the authors speak (a couple of times) of 'Si log'... I suspect that they actually meant 'Si rod'. Please check.

We thank the reviewer for the careful review and have corrected the typographical error.

-on pag.9, the authors should mention that the difference in the number of polariton branches in the two panels of fig.4 is compensated by an additional dark state at the cyclotron frequency. This would also help the readers to understand that the splitting in fig.4(a) originates from the anticrossing of the S-shaped band of fig.4(b) with a state located along the cyclotron line which start mixing with the other modes in going from (b) to (a).

-along similar lines, I might anticipate that adding a third panel to fig.4 showing an intermediate situation could clarify the presentation. As it does not require much extra work, the authors should take this idea in serious consideration.

We thank the reviewer for this helpful suggestion. We have added a third panel to Fig. 4 and modified the captions and the paragraph to clarify this point:

Caption of Fig. 4: “When the cavity mode profiles exhibit high spatial overlap, the splitting vanishes, leading to $N + 1$ polariton branches and an additional dark state along the ω_c line (“coupled scenario”).”

Main text: “When the different cavity modes within the active region highly overlap, these modes efficiently hybridize through the matter. The LP–UP splitting vanishes, producing $N + 1$ polariton branches with an S-shaped middle polariton (MP) and an additional dark state along the CR line (see Fig. 4c for $N = 2$).”

-around pag.12: I wonder why the authors have focused their attention on the ground-state correlations of the form and have not mentioned the anomalous ones. Could there be any interesting new physics in these latter?

We thank the reviewer for this insightful comment. Indeed, we chose to focus on the “normal” ground state correlations and not the anomalous ones. As shown in Fig. R1, the anomalous correlations exhibit features similar to those discussed in the manuscript. Specifically, while intermode correlations for $\sigma = x$ are smaller than the intramode correlations, they are of comparable magnitude for $\sigma = y$.

The primary reason we decided not to include anomalous correlations is that their scaling with the system parameters is not well captured by the figure of merit η . This can be understood by noting that the anomalous correlations depend on both the normal and anomalous Hopfield coefficients in the polariton eigenmodes, whereas the normal correlations depend solely on the anomalous coefficients. Therefore, it is not surprising that anomalous correlations are not entirely governed by the ultrastrong coupling figure of merit, which only impacts the magnitude of the anomalous Hopfield coefficients.

In this context, we agree with the referee that this represents intriguing new physics. We are currently investigating the scaling behavior of the anomalous ground state correlations, which we plan to address in detail in a future publication.

Fig. R1 Anomalous correlations versus magnetic field for polarizations $\sigma = x$ (left) and $\sigma = y$ (right).

-on pag.12, below eq.5: the authors should mention again the values of η for the diagonal coupling together with the ones for the non-diagonal coupling, so that readers can make themselves the comparison of the two cases.

We have included the values of the diagonal terms of $\eta_{pp',\sigma}$ in the text for clarity.

-on pag.13: I do not understand why the parameter Λ in eq.(6) is said to be independent of ω_c . What factor compensates the ω_c in the denominator?

The expression for $\tilde{g}_{p,\sigma}$ in the numerator of Λ_σ includes a $\sqrt{\omega_c}$ term:

$$\tilde{g}_{p,\sigma}(\rho) = g_{p,\sigma,y}(\rho) - ig_{p,\sigma,x}(\rho) = [E_{p,\sigma,y}(\rho, z_{2\text{DEG}}) - iE_{p,\sigma,x}(\rho, z_{2\text{DEG}})] \sqrt{\frac{e^2 \omega_c n_e}{4\epsilon_0 m_{eff} \omega_p a}}.$$

Thus, the ω_c in the denominator of Λ_σ is balanced by the spatial integral $\int d\rho \tilde{g}_{1,\sigma}(\rho) \tilde{g}_{2,\sigma}^*(\rho)$ in the numerator. We have edited the text to clarify that point.

-a couple of lines later, I do not understand why the MP would become a pure-matter mode if it is composed of the two cavity modes. Please check and clarify.

We thank the referee for pointing out this important misprint. We have corrected the sentence, which now reads: “the S-shaped MP would become a hybrid mode equally composed of the two cavity modes”. This correction is clearly illustrated in Fig. S1c of the Supplementary Information.

-The authors should mention in a more explicit way that a similar physics has been very recently investigated in Ref.[25], even though on a completely different physical setup. As the two works were submitted (to the best of my understanding) around the same time, this fact does not reduce the importance of the present work nor raises any concern about scientific priority. Still, readers may take benefit of being pointed to this other interesting work, possibly with some commenting sentences.

We thank the reviewer for the suggestion. We have added a few sentences to discuss the alternative approach reported in Mornhinweg (2024) and highlight the significance of both works.

“Interestingly, a similar phenomenon was recently observed using a different approach in a THz metasurface array [26]. Despite the inherent orthogonality of the cavity mode profiles over the entire unit cell of the resonator, the spatial overlap remains finite when only a subregion of the resonator is considered. To exploit this, Mornhinweg *et al.* patterned the quantum well to confine the active region to the central gap area, where the mode profiles highly overlap. These complementary works highlight the crucial role of spatial profiles in multimode coupling and emphasize the potential of engineered mode profiles and spatial overlap optimization in advancing our understanding of light–matter interaction in exotic regimes.”

RESPONSE TO THE 3RD REVIEWER:

The manuscript presents a study of a 3D photonic crystal in the one-terahertz frequency range, with a planar defect containing a 2D electron gas that reveals so-called ultrastrong coupling with the confined fields. The study combines experiments, computer simulations, and theory, where the experiments are the central contribution.

I have a number of concerns on the paper (numbered as broad categories below), namely (1) on the paper as a whole, notably the broad impact; (2) on the sample; (3) on the experiments; (4) on the simulations.

1) Broad impact:

Let us discuss main features of the manuscript, using keywords from the title and abstract:

A) Multimode: I am not sure what is meant with this keyword, and why "multimode" is so remarkable when every real system is necessary multimode, for instance, always described by a dense set of states (and wave vectors).

We thank the reviewer for this question. While real experimental systems are inherently multimode, previous cavity QED studies in solid-state systems have predominantly been modeled using the single-mode Hopfield model. This approach is justified because the light–matter coupling strength is typically less than the frequency separation between cavity modes, preventing simultaneous strong coupling of multiple modes with the matter.

In contrast, the three-dimensional photonic crystal cavity (3D-PCC) discussed in our manuscript can support several photonic modes within a narrow frequency range. This configuration enables the mixing of photonic modes through the interaction with matter. Moreover, the intricate structure of the 3D-PCC allows tailoring of cavity modes with specific mode profiles, significantly influencing multimode coupling. For example, our findings reveal that cavity modes excited by orthogonally polarized light, even at the same frequency, exhibit distinctly different multimode coupling behaviors when a two-dimensional electron gas (2DEG) is embedded within the cavity. To accurately model these observations, it is essential to extend the standard Hopfield model to incorporate multiple photonic modes and account for the discrete reciprocal lattice vectors of the 3D photonic crystals. This unique multimode coupling behavior, arising from the interaction between photonic modes and matter, remains largely unexplored, particularly in the context of photonic-crystal cavities. Hence, we believe that “multimode ultrastrong coupling” is noteworthy.

B) Ultrastrong coupling: While "strong coupling" has been a topic of considerable interest in cavity QED - albeit accompanied by considerable debate on what is really meant by the notion - it is less clear how "ultrastrong coupling" is defined, and unfortunate that the manuscript does not really explain it.

We thank the reviewer for pointing this out. We added the following sentence to the introduction to clarify the definition of ultrastrong coupling:

“Nevertheless, fabrication challenges in 3D-PCCs have prevented the realization of extreme regimes of light–matter interactions, which necessitate high quality factors and strong electric field confinement [17]. This includes the strong and the ultrastrong coupling [18, 19] (USC) regimes, the latter being achieved when the light–matter coupling strength is a non-negligible fraction of the bare resonance frequencies.”

In addition, the USC regime and its unique features are further discussed in the context of our system on page 12 of the manuscript.

C) Three-dimensional photonic crystal cavity: While I agree that the crystal structures top and bottom have a 3D structure, the central planar defect layer is structurally one-dimensional (1D), so I wonder if the confinement is truly 3D. I am ready to be convinced that the 1D-confined EM waves experience lateral periodicity of both 3D structures, which would induce at most 2D periodicity in the physical

properties. In the community, however, it is generally agreed that 3D cavities entail 3D confinement, which is not demonstrated here.

We appreciate the reviewer's thoughtful comment. While we agree with the reviewer that our planar defect layer only breaks periodicity in one dimension (the z direction), we would like to emphasize that the confinement remains 3D. Indeed, the electric field of the cavity modes we discussed extends into the first two silicon rod layers on both sides of the central planar defect layer along the z direction, as shown in Fig. 2a & b and discussed in detail in section S1.3 of Supplementary Information. These silicon layers, located at different z positions, exhibit alternating periodicity in the x and y directions. The cavity mode profiles result from the combined periodicity in the x and y directions within these silicon rod layers and the broken periodicity in the z direction, reflecting the characteristics of the 3D crystal structure. Consequently, the electric field is confined in all three dimensions, even though the confinement in one of the lateral directions may be weaker at certain z positions. This is further supported by the in-plane spatial profiles of the cavity modes, which are non-uniform in both the x and y directions (Fig. 2c–f).

Alternatively, as noted in the literature [e.g., Solid State Commun. 89, 413–416 (1994)], the infinite woodpile structure without a defect layer exhibits a complete photonic bandgap, which is a signature of a 3D confinement. The inclusion of a defect layer in the structure introduces guided modes within this 3D bandgap, which are thus inherently confined in all three dimensions.

D) Terahertz frequencies: While I applaud challenging experiments as presented by this valiant team of authors, let us agree the range up to 1 THz may hold potential for interesting physics (in the present case the jury "is still out"), but the potential for large-volume device applications is exceedingly slim, since there are no practical devices operating at 1 THz and cryogenic temperatures. One may then argue that since EM is scale-invariant, studies like this one point lead way to optical applications. But this is also not convincing, given that the likelihood of realizing these phenomena at optical frequencies (>100 THz) seem far away.

We appreciate the reviewer's thoughtful comments and the acknowledgement of the challenges involved in our work. We would like to clarify that our study is rooted in fundamental science. Unlike optical frequencies, the THz frequency range holds particular significance in condensed matter physics, as it encompasses a wide variety of low-energy collective excitations, including phonons, magnons, charge density waves, and superconducting gaps. Thus, fabricating devices operating in the THz frequency range is not only more practical than at optical frequencies, due to the larger lattice constants, but also highly compelling for investigating and coupling these collective excitations with cavity photons.

We do not claim that the devices investigated in this study are intended for large-scale applications. Our main focus is on uncovering and understanding the fundamental mechanisms at play in the ultrastrong coupling regime. These insights may serve as inspiration for future developments, even if the path to large-scale applications remains uncertain at this stage.

Finally, we believe that the insights gained from this work can provide valuable perspectives for other frequency regimes, including the optical range. Fundamental science often lays the groundwork for future

technological breakthroughs that may not be immediately foreseeable, and we hope that our findings will contribute to this broader landscape of discovery.

2) Sample:

A) In most photonic crystal studies, considerable attention is paid to characterizing the detailed structure of newly fabricated crystal, and especially of the disorder. So it is unfortunate that this manuscript provides little detail on the realized sample; only a very low-resolution image is in the inset of Figure 1. It is relevant to provide close-up images of the sample both from the top and from the side, including cross-sections, to assess the sample quality. Given the fairly macroscopic character of the sample (layer thickness about 60 to 85 micron) it would be a great start to present optical microscopy (no need for electron microscopy.)

We appreciate the reviewer's suggestion to provide more detailed images of the sample. We have included microscope images of individual patterned silicon wafers and the stacked structure in Supplementary Information Section 3.3 (Fig. S9). Since the patterned region is surrounded by ordinary silicon (as shown in Fig. 1a in the main text), a side-view image was not included because the photonic crystal structure is not visible from this perspective. We hope these additional images help to better convey the quality and structure of the fabricated sample.

B) And complementarily, scattering studies would be very helpful to assess usual distributions that conveniently characterize disorder.

We would like to emphasize that in contrast to devices operating at optical frequencies, where the level of disorder and fabrication imperfections can be comparable to device features (e.g., the photonic crystal lattice constant), photonic crystals operating at THz frequencies, particularly below 0.45 THz (wavelengths $> 666 \mu\text{m}$), are less affected.

As illustrated in Fig. R2a, the transmittance spectra of the 3D photonic crystal without a defect layer exhibit excellent agreement with simulations. These spectra were obtained using time-domain traces truncated at 70 ps and 300 ps for data analysis. As discussed in the main text and Supplementary Information, while the experimental data with a 300 ps time window provides higher frequency resolution, it also exhibits artificial rapid oscillations in the frequency spectra. These oscillations are caused by unwanted back-reflections of the THz pulses collected at longer delays (the Fabry–Pérot effect).

The observed photonic band gap matches well with the simulation result, except for the upper band edge, which appears at a slightly higher frequency in the experimental data. This discrepancy may arise from variations in layer thickness or minor deviations in rod widths during the fabrication process. Importantly, the multimode light–matter coupling discussed in the main text occurs below 0.45 THz, where experimental data and simulations exhibit good agreement.

Figure R2b demonstrates that the transmittance spectra of the 3D photonic crystal without a defect layer are degenerate for orthogonal polarizations, further confirming the high quality of the fabricated structure.

The overall transmission in experiments decreased when a defect layer was introduced into the photonic crystal. This reduction is primarily due to the relatively rough surface of the defect layer after mechanical polishing. However, these imperfections do not change the spectral shape significantly. Further details regarding the transmittance spectra of a bare cavity with a defect layer are discussed on page 13 of this response letter.

Based on the points outlined above, we believe that scattering studies, while valuable in general, are less critical in this specific case. We have added Fig. R2 and the relevant discussion to Supplementary Information Section 3.4.

Fig. R2 THz transmission spectra of a fabricated 3D photonic crystal without a defect layer. a, Transmittance spectra of a woodpile structure with 2 unit cells, shown on both (top) a linear scale and (bottom) a logarithmic scale. The black curve represents the simulation results, while the red and blue curves denote experimental data for polarizations $\sigma = y$ using time windows of 70 ps and 300 ps, respectively. **b,** Transmittance spectra of the 3D photonic crystal for orthogonal polarizations. The bottom panel shows the corresponding logarithmic-scale plot.

C) Finally the symmetry of the layers in the structure is surprising since there is no obvious symmetry. Naively, one would have expected the layers adjacent to the 1D planar defect to be of the same character (i.e. both woodpiles parallel or perpendicular.) If there is a compelling reason why the present choice was made, then it has unfortunately not become clear.

The conventional woodpile crystal is constructed by stacking rod layers in an alternating orthogonal arrangement, where successive layers are oriented perpendicular to each other [Solid State Commun. 89, 413–416 (1994)], as shown in Fig. R3a. The unit cell consists of four layers, with the rods in the third and fourth layers shifted by half of a lattice constant relative to those in the first and second layers, respectively. In our design, we break the periodicity along the z direction by introducing a planar defect layer between two unit cells of the woodpile crystal. As a result, the rod layers on either side of the defect are inherently perpendicular, as shown in Fig. R3b.

Fig. R3 Schematic diagram of the woodpile crystal and its modification with a planar defect. **a**, Structure of the woodpile crystal with two unit cells stacked in the z direction. Each unit cell (blue) consists of four layers of periodic rod, with their orientations alternating between the x and y directions, as indicated by the red labels. The lattice parameter is denoted by a . **b**, Introduction of a planar defect layer (green) between the unit cells. The defect separates the rod layers, with the orientation of the rods in adjacent layers above and below the defect remaining orthogonal.

To establish mirror symmetry along the z direction, one could intentionally invert the stacking order of one of the unit cells. However, we specifically maintain the current design for the following reasons:

(i) **Complete (3D) photonic bandgap**

The (infinite) conventional woodpile structure features a complete photonic band gap filled with guided modes that are induced by the defect layer. The existence of this complete photonic band gap provides those guided modes with high Q factors, which would not be the case with the mirror symmetric structure that lack a complete photonic band gap.

(ii) **Distinct spatial profiles for orthogonal polarizations**

The broken mirror symmetry in the z direction enables distinct spatial profiles for the cavity modes excited by orthogonal polarizations σ ;

(iii) **Sufficiently strong electric field for both polarizations**

The symmetry that combines 4-fold rotational symmetry around the z axis and a mirror symmetry in the z direction ensures that the cavity modes excited by orthogonal polarizations σ are degenerate. Furthermore, both $\sigma = x$ and $\sigma = y$ modes exhibit sufficiently strong electric field at the 2DEG position, enabling the achievement of ultrastrong coupling.

This design allows us to clearly demonstrate the critical role of high-Q cavity mode profiles in multimode coupling, which is the central phenomenon discussed in the manuscript, because the cavity modes excited by orthogonal polarizations exhibit the same resonance frequency and comparable light–matter coupling strengths, with their primary distinction being the spatial profiles. These variations in spatial profiles significantly influence the vacuum Rabi splitting observed in the spectrum due to the hybridization of the cavity modes through their interaction with the matter. For example, our results (Fig. 3 and 5 in the main text) reveal two distinct multimode coupling scenarios, as illustrated in Fig. 4 in the main text. If a woodpile crystal with mirror symmetry were used, the cavity modes excited by orthogonal polarizations would no longer be degenerate, and would feature lower Q factors. Moreover, the coupling strength of the modes for one of the polarizations may be insufficient to reach the ultrastrong coupling regime — or even the strong coupling regime.

We have added a sentence at the bottom of page 3 and a sentence at the bottom of page 5 to further clarify this point:

“Here, we consider the woodpile structure where the rod arrays are stacked in an alternating orthogonal pattern, which is known to exhibit a complete photonic band gap.”;

“Importantly, this structure is crucial in investigating the role of cavity mode profiles in multimode coupling because the cavity modes excited by orthogonal polarizations exhibit the same resonance frequency and comparable light–matter coupling strengths, with their primary distinction being the spatial profiles.”.

3) Experiments:

A) Setup: A description of the setup and experimental conditions is unfortunately missing. This could easily be described in the Supplementary. For instance, while I suspect that the experiments are being conducted at cryogenic temperatures. Similar about the magnetic field dependencies, which are impressive, but unfortunately no description is given how these are maintained (and calibrated).

We thank the reviewer for the comment. We have provided more details in the description of the setup in the Methods section at the end of the main text. We have included the transmission spectra of a QW sample without the cavity in Supplementary Information (section S3.2). The observed CR frequency and the extracted effective mass, which are consistent with previous studies, further confirm the stability of the applied magnetic field.

B) Data: information is missing on essential experimental features that would help to assess the quality of

the experiments. For instance, in both time and frequency domains questions arise: what is the noise floor and how does it depend on frequency, what is the spectral resolution? Etcetera.

We measured the noise floor following the method described in J. Neu & C.A. Schmuttenmaer, J. Appl. Phys. 124, (2018). We first measured the THz signal through an empty aperture. Then, the noise measurement was performed with the THz beam blocked. We repeated the noise measurement three times and took the root-mean-square of the THz time-domain trace. Fig. R4a & b compared the THz signal and the noise level. All traces are normalized to the maximum value of the THz signal. The noise curve was smoothed to avoid spikes, as described in J. Neu (2018). In the frequency range of interest (0.3–0.45 THz), the noise is lower than 10^{-5} in Fig. R4b. Figure R4c shows the dynamic range of the setup, which is defined as the ratio between the signal and noise level. The dynamic range exceeds 50 dB in this frequency range.

Fig. R4 Noise floor analysis of the THz setup. **a**, Power spectra of the THz signal and the root-mean-square noise signal, both normalized to the maximum value of the THz signal. The “noise smoothed” denote the noise spectrum smoothed using a Gaussian filter. **b**, A magnified view of panel (a). **c**, The dynamic range of the THz setup.

The intrinsic spectral resolution of the time-domain data is determined by the length of the time window used in the Fourier transformation: $\Delta f_{\text{intrinsic}} = 1/T$, where T is the time window duration. However, the precision of peak frequency extraction can be improved through zero padding and Lorentzian peak fitting. While zero padding does not improve the actual frequency resolution, it facilitates finer interpolation between frequency-domain data points. Subsequently, curve fitting to the data points allows the peak frequency to be determined with a precision finer than the intrinsic resolution. The estimation of the uncertainty in the extracted peak frequency is discussed on page 17 in this response letter.

We have included a section in Supplementary Information to clarify the experimental features.

C) Transmission: Early in the text, in Figure 1, one would hope to see the measured transmission of the fabricated structure (even better, for reproducibility: multiple structures?) Not only to assess ensuing steps in this optical study, but also to assess the sample quality. Deep in the supplementary, Figure S1 shows the measured transmission (aside: given that the scanning time-resolved measurement method is non-standard, it would be very helpful to see a number of legible raw measurements.) In Figure S1 it is first of all not clear why the simulations are fudged by 60% to match the experiments. Secondly, it is very difficult to distinguish experimental features from simulations; for instance, the sharp peak near 0.48 THz (is it measured or computed?) Thirdly, why do the spectral shapes of apparently important peaks "1" and "2" differ markedly between simulations and experiments? What are the consequences for the claimed cavity-QED behaviour?

As discussed in a previous response, the transmittance spectrum of the photonic crystal without a defect layer (Fig. R2) matches well with the simulation without requiring scaling, demonstrating the high-quality of the photonic crystal structure itself. When a defect layer is introduced, the overall transmission amplitude decreases due to surface roughness and imperfections introduced to the defect layer during mechanical polishing. For this reason, the simulation was scaled by a factor of 0.6 to match the experimental data.

Importantly, these imperfections primarily reduce the transmission amplitude without significantly altering the spectral features. For instance, as shown in Fig. R5, two peaks near the frequencies of modes 1 and 2 in the simulation are observed in the experimental data using a time-domain trace truncated at 70 ps (red trace). As discussed in the main text and Supplementary Information, modes 3 and 4 are not resolved in experiments due to the limited frequency resolution. It should be noted that the actual linewidth of the peaks is narrower, but the intrinsic frequency resolution of the 70 ps data is limited ($\Delta f \sim 0.014$ THz). Extending the time window to 300 ps increases the maximum amplitude of the peaks and decreases their linewidths (blue trace) due to higher intrinsic frequency resolution. However, the spectra obtained with the 300 ps time window also exhibit artificial oscillations caused by back-reflections of the THz pulses. These rapid oscillations result in apparent deviations in the spectral shape of the 300 ps data, which was shown in the original Supplementary Information figure and noted by the reviewer. The spectral shape of the peak corresponding to mode 1 aligns with the simulation result when the oscillations are disregarded. Furthermore, the peak corresponding to mode 2 appears to split due to the influence of these rapid oscillations. Despite these artifacts, the overall spectra features remain consistent with the simulation results, apart from the reduced transmission amplitude. Importantly, the distinct multimode coupling scenarios reported in this work are determined by the magnetic field dependence of the polariton frequencies rather than the linewidths, which are more sensitive to disorder. While small levels of disorder may exist in the system, the magnetic field dependence of polariton frequencies is a more robust parameter, remaining largely unaffected by imperfections. The excellent agreement between the experimentally observed magnetic field dependence of the polariton frequencies and the predictions of our microscopic model underscores the validity of our results.

Regarding the sharp peak near 0.48 THz, it is observed in the simulation and not in the experimental data due to the limited frequency resolution. This peak corresponds to a new cavity mode that redshifts from the upper photonic band, resulting from a thicker defect layer in this case.

Fig. R5 THz transmission spectra of a bare 3D-PCC with an 85- μm -thick defect layer. **a–b**, Transmittance spectra shown **(a)** on a linear scale and **(b)** on a logarithmic scale. The black curve denotes the simulation results scaled by a factor of 0.6 for clarity, while the red and blue curves depict experimental data processed using Fourier transformation with time windows up to 70 ps and 300 ps, respectively. The rapid oscillations in the 300 ps data are artifacts resulting from back-reflections of the THz pulses. The cavity modes, indicated by blue arrows, exhibit lower frequencies compared to Fig. 1 in the main text because the defect layer is thicker in this case. An inset in **(a)** highlights the detailed structure of the low-transmittance region. Modes 3 and 4 are not resolved in the experimental data due to the limited frequency resolution.

To improve clarity and presentation, we have added additional descriptions and revised the figures in Supplementary Information as follows:

1. We now present spectra for both short (70 ps) and long (300 ps) THz time-domain traces. The 70 ps data minimizes the influence of back-reflections of the THz pulses but has limited

frequency resolution. Conversely, the 300 ps data offers higher frequency resolution but exhibits artificial oscillations.

2. The original figure has been divided into two: one comparing the simulation results with the experimental data (Fig. S11), and another comparing the experimental data for two orthogonal polarizations (Fig. S12).

D) Figure S7: with all due respect, the data in all panels are completely unreadable. Why are so many traces plotted? Please show a few representative traces and much enlarged. And while we're at it, please crop the abscissa if interesting features only occupy 1/3rd of the range.

About Figure S7 c): what do "a.u." mean (arbitrary units?) when the scale runs to 1.5×10^{-4} ? And, to appreciate the meaning of the various bursts, it would be helpful to plot amplitude on a semi-log scale (also to appreciate the noise floor) and cropped to discern underlying fringes?

We appreciate the reviewer's feedback and acknowledge that the original presentation of the data in Fig. S7 was overly crowded, making it difficult to discern individual features. To address these concerns and enhance clarity, we have made the following revisions:

1. **Reduced trace density and cropped frequency range [Fig. S13]:**
We retained a figure with multiple traces at different magnetic fields because it is necessary to demonstrate the magnetic-field-dependent peak frequency shifts and the field-independent dips caused by the Fabry–Pérot effect. However, we reduced the number of traces and cropped the frequency range to better highlight the features in each trace.
2. **New logarithmic-scale plot with the noise floor [Fig. S14]:**
To further clarify the data and address the reviewer's suggestion, we added a new figure that presents two representative experimental spectra plotted on a logarithmic scale. This plot also compares the spectra with the noise floor of the setup.
3. **Simplified presentation [Fig. S15]:**
We reduced the number of the traces in Fig. S15c to avoid overcrowding. In addition, we replaced the 7 T data in Fig. S15b with 2 T data, which is also included in the new logarithmic-scale plot in Fig. S14 for consistency.
4. **Clarification of units [Fig. S13c]:**
To address the ambiguity regarding "a.u.", we now explicitly indicate "arb. units" and have normalized the THz electric field data to its maximum value.

We believe these revisions significantly improve the readability of the data.

4) Simulations & presentation:

A) In Figure 1, numerical simulations are shown with an impressive transmittance range, namely from 1 to less than 1×10^{-3} , or ~ 3.5 decades. Since in literature, one rarely sees such high precision, it would be helpful to describe what special measures were taken to arrive at this dynamic range?

We are uncertain about the reviewer's concern regarding the degree of precision of numerical simulations. Based on our experience, such transmission range is not unusual in electromagnetic

simulations, and no special measures were taken to perform the calculations. To verify the robustness of the low-transmittance peaks in the bare cavity simulation, we performed additional simulations using four different mesh sizes (mesh 1–4, Fig. R6). The meshes were defined by controlling the maximum element size in different regions, with mesh 2 corresponding closely to the mesh size used in the manuscript. Mesh 1 represents a coarser mesh size, while meshes 3 and 4 are progressively finer. As shown in Fig. R6b & c, modes 1 and 2 at 338 GHz and 382 GHz remain consistent across all mesh sizes, demonstrating that these peaks, highlighted in Fig. 1 of the main text, are robust. For modes 3 and 4 at higher frequencies with extremely high Q factors, slight variations in amplitude and peak frequency are shown. These differences arise from the lack of full convergence and the 1 GHz frequency resolution, which is insufficient to sample such narrow peaks adequately. For example, mode 3 is represented by only a few data points (Fig. R6d). However, since the multimode coupling discussed in the main text focuses solely on modes 1 and 2, these minor variations in modes 3 and 4 do not affect the conclusions of our study.

Fig. R6 Simulated transmittance spectra as a function of mesh size. **a**, Full transmittance spectra of a bare 3D-PCC with a 60- μm -thick defect layer simulated using varying mesh sizes, ranging from the coarsest (mesh 1) to the finest (mesh 4). **b–d**, Enlarged views of specific frequency ranges from panel (a).

In addition, we simulated the transmittance spectra of a bare 3D-PCC with varying defect layer thicknesses, as shown in Fig. R7. The results show a systematic redshift of modes 1–4 as the defect layer thickness increases. This clear dependence on defect layer thickness further supports the robustness of the peaks observed in the simulations. We have included the plots in Supplementary Information [Fig. S6].

Fig. R7 Simulated transmittance spectra of a bare 3D-PCC as a function of defect layer thickness. The transmittance spectra of a bare 3D-PCC are simulated for defect layer thicknesses ranging from 50 μm to 70 μm . Red arrows mark the shifts of modes 1–4.

B) In Figure 2, it is not clear how the (miniature) symbols are related to the shadings? Including, what are the error bars with all symbols?

We believe the reviewer intended to refer to Fig. 3 rather than Fig. 2, as there are no miniature symbols present in Fig. 2. As discussed in the text, the white dots overlaid on the color plot in Fig. 3 denote the peak frequencies extracted from the transmittance spectra using longer THz time-domain traces for Fourier transformation. The raw time-domain data used to extract the peak frequencies is identical to that used for generating the color plots of the experimental data, except the time-domain data for the color plots are truncated at 33 ps to avoid artificial Fabry–Pérot modulation. The white dots, extracted from long time-domain traces, show agreement with the color plot extracted from short time-domain traces, demonstrating consistency between the data using different time windows.

We have added error bars to the symbols in the updated figure. We identify two primary sources of uncertainty in the extracted peak frequencies. First, the fitting uncertainty at each magnetic field, denoted as Δf_{fit} , is estimated using the 95% confidence interval of the fits. Second, the peak frequency may exhibit slight variations due to system instabilities. These instabilities arise from factors such as laser power fluctuations and minor sample misalignments during measurements. While these variations predominantly affect the transmittance amplitude, the spectral features remain robust, allowing reliable peak frequency extraction. However, slight fluctuations in the extracted peak frequency can still occur.

Fig. R8 Peak frequency variations caused by system fluctuations. a, Transmittance spectra at $B = 0$ T for $\sigma = x$ polarization, measured on different days. A time window of 200 ps was used for the Fourier transformation. **b,** Peak frequencies extracted from the spectra in panel (a).

To evaluate these effects, we plotted the transmittance spectra at 0 T measured on four different days for $\sigma = x$ polarization and compared the extracted UP_1 frequencies (Fig. R8). The standard deviation of these peak frequencies is treated as the systematic error, Δf_{system} . We assume that Δf_{system} remains constant across all values of B . Finally, the total uncertainty in the peak frequency is expressed as $\Delta f_{\text{peak}} =$

$\sqrt{(\Delta f_{\text{fit}})^2 + (\Delta f_{\text{system}})^2}$. The discussion regarding the error bars has been included in Supplementary Information 3.6.

In summary, since the manuscript is unfortunately deficient in many different aspects, publication in Nature Communications is not recommended.

We hope that we have adequately addressed the reviewer's concerns and that she/he will now consider recommending our work for publication in Nature Communications.